# Evaluating and Rewarding Teamwork Using Cooperative Game Abstractions

**Tom Yan**
Carnegie Mellon University
Facebook AI Research
tyyan@cmu.edu

**Christian Kroer**
Columbia University
Facebook Core Data Science
christian.kroer@columbia.edu

**Alexander Peysakhovich**
Facebook AI Research
alexpeys@fb.com

## Abstract

Can we predict how well a team of individuals will perform together? How should individuals be rewarded for their contributions to the team performance? Cooperative game theory gives us a powerful set of tools for answering these questions: the Characteristic Function (CF) and solution concepts like the Shapley Value (SV). There are two major difficulties in applying these techniques to real world problems: first, the CF is rarely given to us and needs to be learned from data. Second, the SV is combinatorial in nature. We introduce a parametric model called cooperative game abstractions (CGAs) for estimating CFs from data. CGAs are easy to learn, readily interpretable, and crucially allow linear-time computation of the SV. We provide identification results and sample complexity bounds for CGA models as well as error bounds in the estimation of the SV using CGAs. We apply our methods to study teams of artificial RL agents as well as real world teams from professional sports.

## 1 Introduction

Suppose we have a group of individuals out of which we need to select a team to perform a task. Besides maximizing team performance, we also wish to reward individuals fairly for their contributions to the team [25]. This general problem arises in many real world contexts: choosing athletes for a sports team [22], choosing workers for a project [28], choosing a subset of classifiers to use in an ensemble [27] etc. In this paper we ask: how can we use data on past performance to figure out which individuals complement each other? How can we then *fairly* compensate team members?

Standard game theory (sometimes called 'non-cooperative' game theory) explicitly specifies actions, players, and utility functions. By contrast, cooperative game theory abstracts away from the 'rules of the game' and simply has as primitives the agents and the characteristic function (henceforth CF). The CF measures how much utility a coalition can create. Solution concepts in cooperative game theory have been developed to be 'fair' divisions of the total utility created by the coalition. These solution concepts can be viewed either as prescriptive (i.e. this is what an individual 'deserves' to get given their contribution) or predictive of what will happen in real world negotiations, where the intuition is that coalitions (or individuals) that don't receive fair compensations will opt to leave the game and simply transact amongst themselves.

These tools are useful for answering our main questions. The CF tells us how well a team will perform and the solution concepts will tell us how to divide value across individuals. For the purposes

of this paper, we consider one of the most prominent solution concepts: the Shapley Value (SV). However, there are two hurdles to overcome.

1. The CF is unknown to us, and is combinatorial in nature, thus requiring a sensible parametric model through which we can learn the CF from team performance data.
2. The SV requires an exponential number of operations to compute.

We introduce the cooperative game abstraction (CGA) model that simultaneously addresses *both* of these issues. In addition, CGA models are interpretable so as to aid analysts in understanding group synergy. Our main idea is motivated by a particular decomposition of the CF into an additive series of weights that capture $m$-way interaction between the $n$ players for $m = 1, ..., n$. When we zero out terms of order order $k + 1$ and higher, this leaves behind an abstraction, a sketched version of the real cooperative game, which we refer to as a $k$th order CGA.

**Our Contribution:** To the best of our knowledge, we are the first to estimate characteristic functions with lossy *abstractions* [12] of the true characteristic function using parametric models, and bound the error of the estimated CF and SV. The second order variant of the CGA was first proposed in [9]. We generalize this work to study CGA models of *any order*. Our theoretical contributions are as follows: (i) sample complexity characterization of when a CGA model of order $k$ (for any order $k$) is identifiable from data (ii) sensitivity analysis of how the estimation error of the characteristic function propagates into the downstream task of estimating the Shapley Value.

Empirically, we first validate the usefulness of CGAs in artificial RL environments, in which we can verify the predictions of CGAs on counterfactual teams. Then, we model real world data from the NBA, for which we do not have ground truth, using CGAs and show that its predictions are consistent with expert knowledge and various metrics of team strength and player value.

## 2 Related Work

Past works on ML for cooperative games have largely been theoretical and focus either on estimating the CF or estimating the Shapley Value directly without the CF. This differs from our goal, which is to model *both* with a provably good model that demonstrates sound performance on real world data. Indeed, our central premise is that the CF is unknown and needs to be learned from data. To the best of our knowledge, we are the first design a compact representation for the CF with *learning from samples in mind*: CGA not only has good learning theoretic properties, but also allows for fast SV computation.

Below, we describe related work in machine learning and cooperative game theory that assume the CF is unknown. In the appendix, we list additional, more distantly related work that assume the CF is known.

**Modeling Characteristic Functions:** As mentioned previously, [9] is the first to consider what we consider the second order variant of CGA. However, its focus was on the computational complexity of the *exact* computation of the Shapley Value. We consider the generalization of this representation to any order and are concerned with using lower rank CGA as an abstraction of complex games for *computational tractability*. As the low rank CGA is a *lossy* estimator of the true CF, we study and obtain theoretical bounds on the estimation errors of what we aim to compute: the CF and the SV.

A related work is [10] which proposes the MGH model for CFs. While the MGH model is like CGA in that both are complete representations, it contains nonlinearity that makes it harder to optimize and *interpret*. More crucially, the MPH model *does not* admit an easy computation of the SV. On the other hand, there are succinct representation models proposed for CFs that do allow the SVs to be readily computed. These are algebraic decision diagrams [1] and MC-nets [14], which represent CFs with a set of logical rules. However, the key drawback is that these models cannot be readily parameterized in tensor form and optimized using modern auto-grad toolkits, unlike the CGA.

Lastly, there has also been work in learning theory [3] that examines conditions under which a characteristic function can be PAC learned from samples. This work is concerned only with the theoretical learnability of the CF (and not the SV) for *certain classes* of cooperative games. By contrast, we study a concrete, parametric model that can approximate the CF of *any cooperative game*, study how approximation noise propagates into the SV and empirically verify that the model obtains good performance on real data.

**Computing the Shapley Value:** There has also been work that directly approximates the Shapley Value, without first learning the CF [4]. This differs from our goal in that we are interested in estimating *both* the Shapley and the CF. The latter is needed for applications such as counterfactual team performance prediction and optimal team formation, as we will demonstrate in the experiments.

**Team Performance Analysis from Data:** We note that all of the work cited above are theoretical and do not test their model on real world data. [22] is one empirical work that does. They model e-Sports team performances using a 2nd order CGA. Our work differs in that i) we generalize their model and study CGA models *any order* to obtain comprehensive sample complexity bounds ii) we are interested in *fair payoff assignment* in addition to team strength. To this end, we show that CGA allows for easy computation of SV and derive noise bounds for the estimated SV.

**Abstraction in Games:** Abstraction is an idea often used in game theory to make the computation of solution concepts such as the Nash Equilibrium (NE) tractable. One can efficiently solve for the NE of a abstracted game and lift the strategy to the original game. In non-cooperative game theory, the relationship between the quality of abstraction and the quality of the lifted strategy with respect to the original game has been heavily studied [18, 16, 17]. Our analysis characterizes the relationship between the abstractions and the solution concept, here being the Shapley Value, for any cooperative game. To the best of our knowledge, our work is the first to apply abstraction for computational tractability in the context of cooperative games.

## 3   Cooperative Game Theory Preliminaries

We begin with definitions in cooperative game theory.

**Definition 1.** *A **cooperative game** is defined by:*

1. *A set of agents $A = \{1, \ldots, n\}$ with generic element $i$*

2. *A characteristic function $v : 2^A \to \mathbb{R}$*

We will refer to a subset of agents $C \in 2^A$ for which $v(C)$ measures how much utility a team $C$ can create and divide amongst themselves. A 'fair division' of this value can be given according to the Shapley Value.

**Definition 2.** *The Shapley Value of an agent $i$ with respect to team $A$ is:*

$$\varphi_i(v) = \sum_{S \subseteq A \setminus i} \frac{|S|!(n - |S| - 1)!}{n!} (v(S \cup \{i\}) - v(S))$$

The Shapley Value is typically justified axiomatically. It is the unique division of total value that satisfies axioms of efficiency (all gains are distributed), symmetry (individuals with equal marginal contribution to all coalition get the same division), linearity (if two games are combined, the new division is the sum of the games' divisions), null player (players with $0$ marginal contribution to any coalition receive $0$ value). The Shapley value has been widely applied in ML, in domains such as cost-division [31, 26], feature importance [24], and data valuation [15] to name a few.

## 4   Cooperative Game Abstractions

### 4.1   Motivation

To model the characteristic function $v$, a natural set of abstractions can be derived from the fact that the characteristic function $v$ can be decomposed into a sum of interaction terms across subsets of agents. In what follows, we will denote abstractions of $v$ as $\hat{v}$.

**Fact 1.** *There exists a set of values $\omega_S$ for each $S = \{i_1, \ldots, i_k\} \subseteq A$ such that any characteristic function can be decomposed into its interaction form where:*

$$v(C) = \sum_{k=1}^{|C|} \sum_{S \in 2_k^C} \omega_S. \tag{1}$$

*where $2_k^C$ is the set of all coalitions of size $k$.*

Note that Fact 1 implies that CGA is a *complete representation*: a CGA model of order $n$ can model *any* set function. Its downside is that it has $2^n$ parameters to be learned from data. We may elect to truncate higher order terms and use an order $k$ CGA model $\hat{v}$ to model $v$ instead:

**Definition 3.** *A kth CGA model is parameterized by weight vector $\omega$, which includes a weight $\omega_C$ for all coalitions $C$ with $|C| \leq k$. The corresponding $v(C)$ is defined as in equation 1.*

A key property of CGA models is that the Shapley Value may be computed from a simple weighted sum of the CGA parameters.

**Fact 2.** *The Shapley Value of an individual $i$ with respect to players $A$ may be expressed as:*

$$\varphi_i(v) = \sum_{T \subseteq A \setminus \{i\}} \frac{1}{|T|+1} \omega_{T \cup \{i\}}$$

## 4.2  Learning a CGA

We learn the CGA model from samples of coalition values from $v$. Given hypothesis class be $\mathcal{H}$, we perform empirical risk minimization (ERM) with criterion: $\min_{\hat{v} \in \mathcal{H}} \sum_{(C, v(C)) \in \mathcal{D}_P} (\hat{v}(C) - v(C))^2$

An important question that immediately follows is: when can a CGA model be identified from data? We define an exact identification notion as below:

**Definition 4.** *Suppose that we have a set of hypotheses $\mathcal{H}$ from which we will choose $\hat{v}$ via minimization of the criterion above. Suppose the dataset $\mathcal{D}_P$ is actually generated via a true $v^* \in \mathcal{H}$, we say that $\mathcal{D}_P$ identifies $v^*$ if $v^*$ is the unique minimizer of the criterion.*

Identification is an important question for three reasons. We are interested in the parameters of the model since they will be used to (i) predict the performance of unseen teams (ii) compute the Shapley value (iii) understand complementarity and substitutability between team members. If there are multiple sets of parameters consistent with the data, then none of the inferences we perform to answer those questions (e.g the marginal contribution of players) will be well defined.

We now give some sufficiency and necessity conditions on $\mathcal{D}_P$ for $v^*$ to be identified exactly. These results generalize known sample complexity bounds from [29], expanding their bounds for only order 2 to *any* order $k$.

**Theorem 1** (Sufficiency for Identification). *Suppose $\mathcal{H}$ includes all $k^{th}$ order CGAs and $v^*$ is a $k^{th}$ order CGA. If $\mathcal{D}_P$ include performances from all teams of at least $k$ different sizes $s_1, \ldots, s_k \in [k, n-1]$, then $\mathcal{D}_P$ identifies $v^*$.*

**Theorem 2** (Necessity for Identification). *Suppose $\mathcal{H}$ includes all $k^{th}$ order CGAs and $v^*$ is a $k^{th}$ order CGA. If $\mathcal{D}_P$ contains performances of teams of only $m < k$ different sizes, then $\mathcal{D}_P$ does not always identify $v^*$.*

We relegate the full proofs to the Appendix, as they are quite involved. To provide some intuition for the proof, in the sufficiency result, the argument uses induction to exploit structure in the matrix to arrive at the conditions under which its null space is empty, which implies that the matrix is full rank and the CGA is identifiable. In the complementary necessity result, we offer counterexamples that show even with all subsets of $m < k$ different sizes, the matrix corresponding to the system of linear equations may not be full rank, thus making the CGA model non-identifiable from data.

These results show that if the order $k = O(1)$, identification is possible with $\text{poly}(n)$ samples and that if $k = O(n)$, the number of samples becomes exponential in $n$. Therefore, we suggest that practitioners should focus on the lowest order CGAs that they believe are suitable. For us, we find that low rank, second order CGAs demonstrate good performance in our experiments.

One consideration is that these bounds may be too pessimistic in requiring *exact* recovery of the true $v$. In the appendix, we provide sample bounds for identification of CGA under a PAC/PMAC [2] framework (Proposition 1). In particular, we have that under the looser, PMAC approximation notion, only $O(n)$ instead of $O(d_k)$ samples are needed for approximate estimation of most coalition values.

# 5 Approximate Shapley Values

With our approximation of the CF $\hat{v}$ in hand, we examine the fidelity of the SV computed from $\hat{v}$. We denote the approximated SV of player $i$ as $\varphi_i(\hat{v})$ and the real SV $\varphi_i(v)$. As is typical in sensitivity analysis, we derive bounds relating the error in $v$ to the error in the Shapley value.

These bounds may be of independent interest since often in ML applications $v$ is stochastic. For instance, SV is widely used in interpretability literature [7, 8, 24, 6, 11], where $v$ is taken to be the model performance. The model performance is typically stochastic, since it is a function of the random samples of data used to train the model and the randomness in the optimization, which can converge to differing local optima due to the nonconvexity of the losses e.g of deep models.

Let $\varphi(v)$ be the vector of Shapley Values. We start with a worst-case error bound for $\ell_2$ when the adversary can choose how to distribute a fixed amount of error into $v$ to construct $\hat{v}$.

**Theorem 3.** *The $\ell_2$ norm of the estimation error of the Shapley Values is bounded by:*

$$\|\varphi(v) - \varphi(\hat{v})\|_2^2 \leq \frac{2}{n}\|v - \hat{v}\|_2^2 \tag{2}$$

Though this result is *tight*, it assumes a non-smooth, adversarial distribution that places infinite density on the eigenvector corresponding to the largest singular value of the SV operator. Below, we consider average case bounds assuming that the error is of fixed norm and drawn from a *smooth* distribution; this type of assumption is often used in smooth analysis [13].

**Theorem 4.** *Assuming that $v - \hat{v}$ is drawn from distribution $\mathcal{D}_{B_r}$ with support equal to a sphere and smooth in that $\kappa_0 \leq \mathrm{Pr}_{\mathcal{D}_{B_r}}(x) \leq \kappa_1$ for any point $x$ in its support, then:*

$$\mathbb{E}_{v-\hat{v}\sim\mathcal{D}_{B_r}}[\|\varphi(v) - \varphi(\hat{v})\|_2^2] \leq \frac{6}{n}\frac{\kappa_1}{\kappa_0}\frac{\|v - \hat{v}\|_2^2}{2^n} \tag{3}$$

We can generalize these results to any noise distribution thus:

**Corollary 1.** *Suppose noise $v - \hat{v} \sim \mathcal{D}_n$ is such that its conditional distribution satisfies $\kappa_0(r) \leq \mathrm{Pr}_{\mathcal{D}_n}(x|\|x\|_2^2 = r^2) \leq \kappa_1(r)$ for all $r$ and $x$ in $\mathcal{D}_n$'s support, then:*

$$\mathbb{E}_{v-\hat{v}\sim\mathcal{D}_n}[\|\varphi(v) - \varphi(\hat{v})\|_2^2] \leq \frac{6}{n}\mathbb{E}_r\left[\frac{\kappa_1(r)}{\kappa_0(r)}\left(\frac{r^2}{2^n}\right)\right]$$

Intuitively, this means that if the error $v - \hat{v}$ is on average spread out in that $\mathcal{D}_n$ is fairly smooth in expectation across concentric spheres in its support, then the $\ell_2$ error of the Shapley value is small on average. Indeed, an astute reader may worry that only a $\frac{2}{n}$ reduction in the *aggregate* approximation error $\|v - \hat{v}\|_2^2$ is not large enough since $v - \hat{v} \in \mathbb{R}^{2^n}$. Theorem 4 and Corollary 1 show that the SV actually induces a $\frac{6}{n}\frac{\kappa_1}{\kappa_0}$ scaling of the *average* approximation error.

We also obtain analogous worst and average-case $\ell_1$ bounds with scaling factors on the same order. Due to space constraints, please see Theorem 5 and 6 in the appendix for the results.

Lastly, we note that these bounds are general. In the appendix, we obtain a simple derivation of the CGA-specific bias, which can be plugged into these bounds for the SV bias. Note that bias in the estimation of the CF only arises due to model misspecification, i.e if order $k$ is used to model a game of order $r$ for $r > k$. This description covers *all cases* as any CF of a game necessarily corresponds to a CGA model of a certain order (Fact 1) and estimation error only arises due to a smaller order being specified. Certainly, we note that more refined bounds are a natural future extension to this work.

# 6 Experiments

## 6.1 Virtual Teams

We generate team performance data from the OpenAI particle environment [23] [1]. The task in this environment is team-based and requires cooperation: 3 agents are placed in a map and 3 landmarks

are marked, and agents have a limited amount of time to reach the landmarks and are scored according to the minimum distance of any agent to any landmark. In addition, negative rewards are incurred for colliding with other agents. Thus, a team which can cooperate well is able to assign a single landmark per agent in real time and spread out to cover them without colliding with each other.

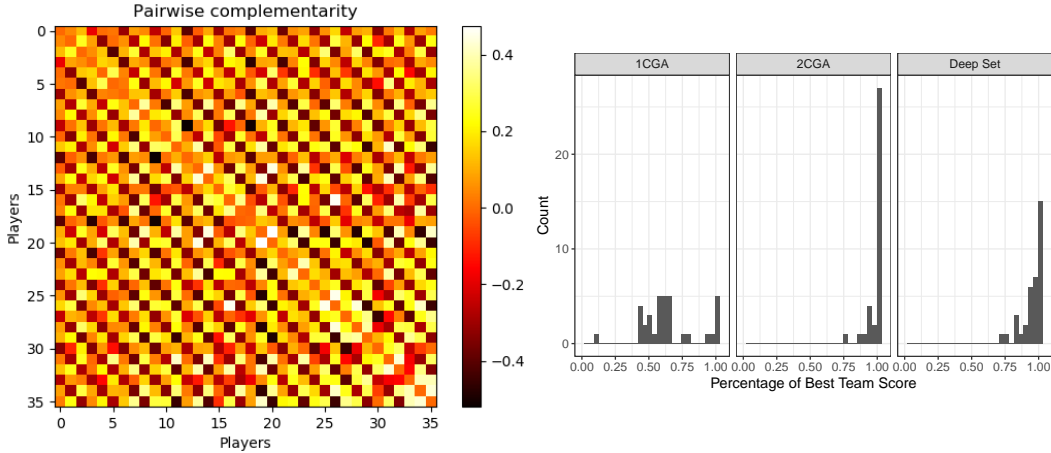

Figure 1: Left: Interaction matrix from second order CGA model. Players are clustered by original training team ($\{0, 1, 2\}$ trained together as did $\{3, 4, 5\}$, etc...). We see complex patterns of complementarity and substitutability as well as a clear replication of the well known fact that agents that train together can coordinate much better than agents which are trained separately - this can be seen in the figure by the strong complementarity in the diagonal blocks of size 3 compared to other $3 \times 3$ off-diagonal blocks. Right: Histograms of ratios of the score attained by the completed team chosen by the model normalized by the score of the actual best team containing a given initial agent.

We train 12 teams of agents (36 agents total) using the default algorithm and parameters from the OpenAI GitHub repo. We then evaluate all $\binom{36}{3} = 7140$ mixed teams of these agents evaluated over 100000 episodes. The train/validation/test split is 50/10/40. We fit a baseline first order (where team = sum of members) CGA and a second order CGA to predicting the final score of each team. We also compare to a more general, state of the art model for learning set functions, DeepSet [32], which is designed purely for prediction.

Since DeepSet contains more parameters than the CGA model, we expect it to fit data better. However, unlike the CGA model, Deepset is (i) less easily identified due to the larger sample complexity needed (ii) *not readily interpretable* due to the non-linearity of $\phi$ (iii) and importantly, one cannot readily compute or estimate the Shapely. To compute the Shapley values exactly, one would have to first compute $v(C)$ for each coalition $C$, thus requiring $2^n$ feed-foward passes through the network. Even to approximate the Shapley value, it is known that $O(n \log n)$ evaluations of the model (network) are needed [15]. In contrast, to compute the Shapley with CGA, only one weighted sum of the CGA model parameters is needed and thus takes $O(1)$ number of evaluation.

**Prediction**: The first order CGA model achieves an test set MSE of of .79, the second order model achieves an order of magnitude smaller at .07. These results show that in this environment teams are not just sums of their parts. The DeepSet model achieves an MSE of .042, showing that we give up some predictive accuracy (but not that much) from using the simpler $2^{nd}$ order CGA. We emphasize that the goal of this experiment is *not* to find the most predictive model. Rather, it is to show that the much smaller, second order CGA model is roughly comparable to Deepset, all the while conferring the advantages of: 1. being interpretable 2. allowing easy computation of the Shapley Value.

**Interpretability:** To the first point, we visualize the learned matrix $\widehat{V}$ of the second order CGA in a heatmap (Figure 1) that allows us to discern players that complement/substitute each other.

**Best Team Formation:** For each of the 36 agents we have trained, we ask: what is the best set of 2 agents to add to them to make a team? More generally, this problem of optimal player addition is one often faced by real world sports teams, as they choose new players to draft or sign so as to further

bolster their team performance. In this virtual setting, we can evaluate all possible additions to the team so as to gauge the predictive performance of our models.

In our setup, we restrict only to possible teammates which the original agent was not trained with. Figure 1 shows the histogram of ratios of the score attained by the completed team, which was selected by the model, normalized by the score of the actual best team. While the first-order CGA fails to construct good teams (since it does not consider any complementarities), the second order CGA and DeepSet model achieve more than $\sim 95\%$ of the possible value. Thus, the complementarity patterns learned via the 2nd order CGA are, in fact, important for this task. We also note that the second order CGA model outperforms DeepSet on this task.

## 6.2 Real World Sports Teams

We now consider a more complex, real world problem: predicting team performance in the NBA. We collect the last 6 seasons of NBA games (a total of 7380 games) from Kaggle along with the publicly available box scores [2]. Unlike in the dataset above, we do not observe absolute team performance, rather we only observe relative performance (who wins). We model matchup outcomes using the Bradley–Terry model. In particular, given the team strengths, the probability of team $i$ winning in a match against team $j$ as:

$$\Pr[w = 1 \mid \hat{v}, C_i, C_j] = \frac{\exp(\hat{v}(C_i))}{\exp(\hat{v}(C_i)) + \exp(\hat{v}(C_j))}$$

This gives us a well defined negative log likelihood (NLL) criterion of the data $\mathcal{D}$, which we optimize with respect to $\hat{v}$. We set each team in each game to be represented by its starting lineup (5 individuals). Then we learn $\hat{v}$ such that it minimizes the negative log likelihood using standard batch SGD with learning rate $0.001$. Because basketball teams are of a fixed size (only one set of sizes), we use L2 regularization to choose one among the many possible set of models parameters.

As with the RL experiment above we compare a first order CGA, a second order CGA, and a DeepSet model. We split the dataset randomly into 80 percent training, 10 percent validation, and 10 percent test subsets. We set hyperparameters by optimizing the loss on the validation set.

### 6.2.1 Results

**Prediction:** How well does the CGA perform in this task? We begin by studying an imperfect metric: out-of-sample predictive performance. First, we see that the NBA performance can be fit fairly well with just a first order CGA - that is, we can think of most teams roughly as the sum of their parts. The first order CGA yields an out of sample mean negative log likelihood of $-.631$ which is slightly improved to $-.627$ under the second order CGA. We do also experiment with a third order CGA which did not improve over the second order CGA performance. This suggests that the second order is an apt choice for the abstraction. Finally, we observe that the DeepSet model is not able to outperform the CGA yielding an out of sample mean NLL of $-.63$.

Overall, we find that predictive accuracy is low, at only about $\sim 65\%$, as a result of the league being very competitive and teams being fairly evenly matched. Thus, predictive accuracy does not tell the whole story and is not the focus of the experiment. Note that the data at hand is observational and while players do move across teams and starting lineups change due to factors such as injuries, time in the season, etc... who plays with whom is highly correlated across years and starting lineups are endogenous (for example, a coach may not start one of their best players when playing a much weaker team to avoid risking injury). Thus, we cannot evaluate counterfactual teams. Instead, we supplement our the predictive analysis with analyses of the competing models to see if they are truly able to extract insights from the data consistent with NBA analytics experts.

**Unseen teams:** We consider teams the model has not seen: NBA All Star teams. During each season, fans and professional analysts vote to select 'superstar' teams of players that then play each other in an exhibition game, which is not included in our training data. We collect every All Star team from the time period spanning our training set and compare our second-order CGA model scores given to All Star teams with those of 1000 randomly generated, 'average' teams.

Recall that in the matchup datasets, the difference in scores between two teams is reflective of the probability that one team will win in a matchup. Thus, there is no natural zero point like when we are predicting $v$ directly and we have chosen one particular normalization where the average score is zero. If our model does generalize well, it should predict that these all star teams are far above average despite never seeing this combination of players in the training set.

We also investigate whether the CGA has learned things about whole teams (e.g "the Cavaliers usually win") and whether there is sufficient variation in starting lineups that we have learned the disentangled contributions of individual players (e.g the team's success is largely due to Lebron's brilliance). We investigate this by constructing synthetic 'same-team-All-Star' teams where we replace each player in a real All Star team with a randomly selected teammate from their real NBA team from that year.

Figure 2 shows the distribution of scores for randomly constructed teams with red lines representing predicted scores for the real All Star teams and blue lines for predicted scores for the 'same-team-All-Star' teams. These results show that the predictive performance of the CGA in win rate prediction comes from meaningful player-level assessment, not just that certain teams usually win (or lose).

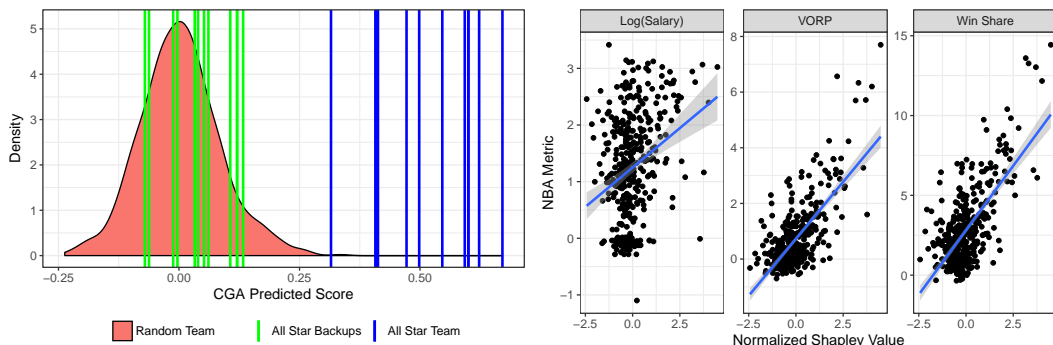

Figure 2: Left Panel: The second order CGA predicts that All Star Teams are far above the $99^{th}$ percentile of random teams. Replacing each All Star with their team-level replacement gives much worse teams. These results show that the predictive performance of the CGA in win rate prediction comes from player-level assessment, and not just memorization of certain teams usually wining or losing. Right Panel: Marginal contributions of individual NBA players, as measured by the Shapley Value from the second order CGA, correlate well with measures of player-level value add used by NBA analysts (VORP, Win-Share) as well as market-level value-add (salary).

**Shapley Value as Individual Measure:** So far we have asked whether our CGA captures team-level performance. We now turn to asking whether it captures individual-level marginal contribution. For each team, we compute the team members' Shapley Values *with respect to that team*. Since our dataset contains multiple years and individuals move across teams, we average an individual's computed Shapley values across all his teams. We correlate the Shapley Value based contribution scores with real world metrics used to evaluate basketball players' marginal contributions. We consider 3 measures commonly used in NBA analytics.

First, we look at the value-over-replacement metric player[3] (VORP). In basketball analytics, VORP tries to compute what would happen if the player were to be removed from the team and replaced by a random player in their position. Second, we look at win-share[4] (WS). Win-share tries to associate what percent of a team's performance can be attributed to a particular player. Finally, we use individual salaries, which are market measures of individual value add. Of course, a players' salary reflects much more than an individuals' contribution to team wins and losses (e.g their popularity, scarcity, etc...) and is extremely right tailed in the case of the NBA, so we consider its log. For each of these metrics, for each player, we average their values across the same years as our dataset.

Figure 2 plots CGA Shapley values against these measures. We see that there is a strong positive relationship between the CGA predicted Shapley value and other measures of individual contribution. Taken together, these results suggest that CGA indeed learns meaningful individual-level contribution measures, in a way that is consistent with expert knowledge.

**Remark:** overall, our experiments highlight the computational benefit of CGAs. In many cases like the NBA, team sizes are small relative to the number of players. We show that this structural prior can be encoded in a low rank CGA model, which does just as well (or better) than more complex, agnostic estimators like DeepSet (our main baseline), and is also interpretable to the benefit of users.

## 7 Conclusion

Cooperative game theory is a powerful set of tools. However, the CF is combinatorial and computing solution concepts like the Shapley is difficult. We introduce CGAs as a scalable, interpretable model for approximating the CF, and easily computing the SV. We provide a bevy of theoretical and empirical results so as to guide the application of CGA to model real world data.

Non-cooperative Game Theory has received much attention from the Machine Learning and AI community [30, 20, 19, 21, 5], while Cooperative Game Theory has been less explored. We believe that the intersection of Machine Learning and Cooperative Game Theory is rich with topics ranging from Multi-agent RL to Federated Learning. Our broader hope is that our work provides a springboard for future research in this area.

## 8 Broader Impact

In this work, we introduce and analyze a general model for team strength and player value. While our end goal is to ensure accurate assessment of team strength, and as a result *fair* distribution of team value, there is the risk of model misspecification and resultant bias in the estimators. Furthermore, often times the team performance data we see is observational and the data we observe may be biased due to individuals being of disparate background. Indeed, accounting for such confounding factors is an important extension to our work that we would like to highlight.

## Acknowledgments and Disclosure of Funding

We wish to acknowledge financial support from the National Science Foundation Graduate Research Fellowship Program. Alex Peysakhovich and Christian Kroer were employed by Facebook Inc. and Tom Yan was a research intern at Facebook AI Research when this research was conducted.

## Footnotes

[1] https://github.com/openai/multiagent-particle-envs

[2]https://www.kaggle.com/drgilermo/nba-players-stats

[3] https://www.basketball-reference.com/leaders/vorp_career.html

[4] https://www.basketball-reference.com/about/ws.html

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
