[Supplementary Material]

# References

[1] Yoram Bachrach, Evangelos Markakis, Ariel D Procaccia, Jeffrey S Rosenschein, and Amin Saberi. Approximating power indices. In *Proceedings of the 7th international joint conference on Autonomous agents and multiagent systems-Volume 2*, pages 943–950, 2008.

[2] R Iris Bahar, Erica A Frohm, Charles M Gaona, Gary D Hachtel, Enrico Macii, Abelardo Pardo, and Fabio Somenzi. Algebric decision diagrams and their applications. *Formal methods in system design*, 10(2-3):171–206, 1997.

[3] Maria-Florina Balcan and Nicholas JA Harvey. Learning submodular functions. In *Proceedings of the forty-third annual ACM symposium on Theory of computing*, pages 793–802, 2011.

[4] Maria Florina Balcan, Ariel D Procaccia, and Yair Zick. Learning cooperative games. In *Twenty-Fourth International Joint Conference on Artificial Intelligence*, 2015.

[5] Eric Balkanski, Umar Syed, and Sergei Vassilvitskii. Statistical cost sharing. In *Advances in Neural Information Processing Systems*, pages 6221–6230, 2017.

[6] Sylvain Béal, Mihai Manea, Eric Rémila, and Philippe Solal. Games with identical shapley values. *Handbook of the Shapley Value*, pages 93–110, 2019.

[7] Noam Brown and Tuomas Sandholm. Superhuman ai for heads-up no-limit poker: Libratus beats top professionals. *Science*, 359(6374):418–424, 2018.

[8] Georgios Chalkiadakis, Edith Elkind, and Michael Wooldridge. Computational aspects of cooperative game theory. *Synthesis Lectures on Artificial Intelligence and Machine Learning*, 5(6):1–168, 2011.

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

[32] Sasan Maleki, Long Tran-Thanh, Greg Hines, Talal Rahwan, and Alex Rogers. Bounding the estimation error of sampling-based shapley value approximation. *arXiv preprint arXiv:1306.4265*, 2013.

[33] Tomasz P Michalak, Karthik V Aadithya, Piotr L Szczepanski, Balaraman Ravindran, and Nicholas R Jennings. Efficient computation of the shapley value for game-theoretic network centrality. *Journal of Artificial Intelligence Research*, 46:607–650, 2013.

[34] Hervé Moulin. *Fair division and collective welfare*. MIT press, 2004.

[35] Ta Duy Nguyen and Yair Zick. Resource based cooperative games: Optimization, fairness and stability. In *International Symposium on Algorithmic Game Theory*, pages 239–244. Springer, 2018.

[36] Lior Rokach. Ensemble-based classifiers. *Artificial Intelligence Review*, 33(1-2):1–39, 2010.

[37] Eduardo Salas, Dana E Sims, and C Shawn Burke. Is there a ?big five? in teamwork? *Small group research*, 36(5):555–599, 2005.

[38] Arjun Seshadri, Alexander Peysakhovich, and Johan Ugander. Discovering context effects from raw choice data. *ICML 2019*, 2019.

[39] Shai Shalev-Shwartz and Shai Ben-David. *Understanding machine learning: From theory to algorithms*. Cambridge university press, 2014.

[40] Yoav Shoham, Rob Powers, and Trond Grenager. If multi-agent learning is the answer, what is the question? *Artificial Intelligence*, 171(7):365–377, 2007.

[41] X Tan and TT Lie. Application of the shapley value on transmission cost allocation in the competitive power market environment. *IEE Proceedings-Generation, Transmission and Distribution*, 149(1):15–20, 2002.

[42] Tom Yan and Ariel D Procaccia. If you like shapley then you'll love the core.

[43] Manzil Zaheer, Satwik Kottur, Siamak Ravanbakhsh, Barnabas Poczos, Ruslan R Salakhutdinov, and Alexander J Smola. Deep sets. In *Advances in neural information processing systems*, pages 3391–3401, 2017.

# A  Appendix to "Evaluating and Rewarding Teamwork Using Cooperative Game Abstractions"

## A.1  Identification Theorem Proofs

**Theorem 1** (Sufficiency for Identification). *Suppose $\mathcal{H}$ includes all $k^{th}$ order CGAs and $v^*$ is a $k^{th}$ order CGA. If $\mathcal{D}_P$ include performances from all teams of at least $k$ different subset sizes $s_1, \ldots, s_k \in [k, n-1]$, then $\mathcal{D}_P$ identifies $v^*$.*

*Proof.* Let $\boldsymbol{w}$ be the first-through-$k$'th order weights we seek to learn, with the first $n$ indices corresponding to $\omega_S$ such that $|S| = 1$, the next $\binom{n}{2}$ indices corresponding to $|S| = 2$, and so on up through the last $\binom{n}{k}$ terms corresponding to $|S| = k$. Let $\boldsymbol{v}$ be the corresponding coalitional values we observe.

Finding a $k$'th-order CGA corresponding to $\mathcal{D}$ can be formulated as finding a solution to $M\boldsymbol{w} = \boldsymbol{v}$, where matrix $M$ is a matrix whose rows correspond to the data points and each entry in the matrix $\in \{0, 1\}$. For a given datapoint $(S, v(S))$, the corresponding row has ones in all entries corresponding to interaction terms $\omega_T$ such that $T \subseteq S$. Note that we only consider subset sizes $\geq k$, since subsets sizes smaller than $k$ would not exhibit $k$th order interaction.

To show identifiability, it suffices to show that $M$ has rank equal to the column size, since otherwise the null space is non-empty and there exist multiple $\boldsymbol{w}$ which satisfies the equation. Equivalently, a full rank matrix ensures that the optimization criterion is strictly convex and that the minimizer is unique. Define matrix $M_{ntk}$ to be the submatrix consisting of all rows from all subsets of size $t$ and columns corresponding only to that of the $k$'th order weights.

$$
M = \begin{array}{c} \text{rows from subsets of size } s_1 \\ \ldots \\ \text{rows from subsets of size } s_k \end{array} \overset{\begin{array}{ccc} \text{first order weight} & \ldots & k\text{'th order weight} \end{array}}{\left( \begin{array}{ccc} M_{ns_1 1} & \ldots & M_{ns_1 k} \\ \ldots & \ldots & \ldots \\ M_{ns_k 1} & \ldots & M_{ns_k k} \end{array} \right)}
$$

We will now show that we can perform row reductions on the decomposition into submatrices, such that we end up with all zeroes below the antidiagonal.

First we note that every row in $M_{nbk}$ is a linear combination of rows in $M_{nak}$ for any $a < b$. Consider a row $s_b$ corresponding to subset $\{i_1, ..., i_b\}$. We take all the rows in $M_{nak}$ corresponding to subsets $s'$ where $s' \subseteq \{i_1, ..., i_b\}$ and $|s'| = a$. We sum all $\binom{b}{a}$ of these rows, and denote this row $s'_b$. Looking at a particular $k$th order weight, say WLOG corresponding to $\{i_1, ..., i_k\} \subseteq \{i_1, ..., i_b\}$, there is a 1 in this column in $s_b$. This subset of size $k$ shows up in $\binom{b-k}{a-k}$ subsets of size $a$. Therefore, the corresponding entry in row $s'_b$ is $\binom{b-k}{a-k}$. And so, we can derive that $\binom{b-k}{a-k}^{-1} s'_b = s_b$ as they both have the same support: every subset of size $k$ in $\{i_1, ..., i_b\}$ can be found in a subset of $\{i_1, ..., i_b\}$ of size $a$.

Thus we may use an appropriate multiple $\alpha$ of the first row to replace $M_{ns_i k}$ with a zero for any $i > 1$. However, this changes the whole row, and so the $j$'th order term changes to $M_{ns_i j} - \alpha M_{ns_1 j}$. But by the same logic as for $k$, summing the $l$'th weights gives the same row scaled by $\binom{b-l}{a-l}$, and thus $M_{ns_i j} - \alpha M_{ns_1 j} = \left( 1 - \frac{\binom{b-l}{a-l}}{\binom{b-k}{a-k}} \right) M_{ns_i j}$.

The above shows that we can perform row reduction using the first row of submatrices in order to put zeroes in the last column while retaining all submatrices in other columns (up to rescaling). But now we may apply this logic inductively, by considering only the submatrices corresponding to first through $k-1$'th order weights and rows from subsets of size $s_2$ or greater, and so on. We get that the matrix $M$ looks as follows after row reduction:

|  | first order weight | ... | (k-1)th order weight | kth order weight |
|---|---|---|---|---|
| rows from subsets of size $s_1$ | $M_{ns_11}$ | ... | $M_{ns_1(k-1)}$ | $M_{ns_1k}$ |
| rows from subsets of size $s_2$ | $M'_{ns_21}$ | ... | $M_{ns_2(k-1)}$ | 0 |
| ... | ... | ... | 0 | 0 |
| rows from subsets of size $s_k$ | $M_{ns_k1}$ | ... | 0 | 0 |

where $M'_{ns_21}$ denotes submatrices above the antidiagonal that have been rescaled (note that the first row does not need rescaling). It is then sufficient to show that $M_{ns_1k}, M_{ns_2(k-1)}..., M_{ns_k1}$ (note that each of these submatrices has more rows than columns) are all full rank to show $M$ is full rank.

To do this, we first prove a lemma.

**Lemma 1.** *When $t \geq k$ and $n = t + k$, the matrix $M_{ntk}$ is full rank.*

*Proof.* We will proceed by induction on $k$.

*Base case ($k = 1$):* Since $k = 1$ each row corresponds to an all-one row with a single zero for the agent left out. Since we have such a row for each agent that can be left out, we get $n$ linearly-independent rows.

Thus the matrix is full rank.

*Induction step ($k > 1$):*

Assuming this statement holds for orders $1, \ldots, k - 1$. We will prove the statement for when the order is $k$. To do this we will use induction on $n$:

*Base case ($n = 2k$):* $n = 2k \Rightarrow t = k$, and so $M_{ntk}$ is the identity matrix and is thus full rank.

*Induction step ($n > 2k$):* Assume the matrix is full rank for when number of players is equal to $2k, ..., n$. To prove the matrix is full rank for $n + 1$, consider the following decomposition of $M_{(n+1)(n+1-k)k}$.

|  | weights of subsets including 1 | weights of excluding including 1 |
|---|---|---|
| subsets including 1 | $A$ | $B$ |
| subsets excluding 1 | 0 | $C$ |

Observe that matrix $B$ corresponds to $M_{n(n-k)k}$ and is thus full rank by induction hypothesis. This means we can use linear combinations of rows of $B$ to reduce rows in $C$. In particular, we can performs row reductions such that we replace $C$ with zeroes: For each row in $C$ corresponding to a team of size of $n + 1 - k$ selected from $[2, ..., n]$, we consider all $n - k$ subsets of this team in $B$ and sum them. For any subset of this team of size $k$, then we see that it shows up in the sum: $\binom{n+1-k-(k)}{n-k-(k)} = n + 1 - 2k$ times. Therefore, the sum is $n + 1 - 2k$ times the row in $C$.

Moreover, let $D$ be the $n + 1 - k$ rows from $A$ summed together when performing the row reduction, let the resultant matrix be $D$. The reduced matrix looks like:

$$M_{(n+1)(n+1-k)k} =$$

|  | weights of subsets including 1 | weights of excluding including 1 |
|---|---|---|
| subsets including 1 | $A$ | $B$ |
| subsets excluding 1 | $D$ | 0 |

Then, we observe that $D$ corresponds to a scaled version of $M_{n(n+1-k)(k-1)}$, which is full rank by the inductive assumption. The scaling factor is calculated as follows: For a subset $\{1, ..., k\}$, the $k - 1$ elements show up in the $n + 1 - k$ subset row of $C$, then shows up in $\binom{n+1-k-(k-1)}{n-k-(k-1)} = n + 2 - 2k$ of the $n - k$ subsets. And so, $D$ is a $-\frac{n+2-2k}{n+1-2k}$ scaled version of $M_{n(n+1-k)(k-1)}$.

$B$ remains unchanged after the row reduction and is full rank and thus the whole matrix is full rank. $\square$

With this lemma in hand we can return to the main proof. To complete the proof, we will show that for any $k$, any $n \geq 2k$ and any $t \in [k, n-k]$, $M_{ntk}$ is full rank.

Note that $t \in [k, n-k] \Rightarrow \binom{n}{t} \geq \binom{n}{k}$ (as otherwise number of rows is already fewer than number of columns and the matrix will have rank less than column size).

We will use induction on $k$.

*Base case ($k = 1$)*: by Lemma 1, the matrix is full rank when $k = 1$ for any team size $t$ and number of players $n$.

*Induction step ($k > 1$)*: assumes this holds for orders $1, ..., k-1$ and any $t$ and $n$. For order $k$, fix some $t \geq k$, we will show the matrix is full rank for all $n \geq t + k$ by induction on $n$. For the base case $n = t + k$ the matrix is full rank by Lemma 1. For the induction step assume $n > t + k$: assume the matrix is full rank when the number of players is in $\{t+k, ..., n-1\}$. Now when the number of players is $n$, we may decompose the matrix into columns corresponding to weights of $k$-size subsets containing player 1, and rows into teams including or excluding player 1.

$$M_{ntk} = \begin{array}{c} \\ \text{subsets including 1} \\ \text{subsets excluding 1} \end{array} \overset{\begin{array}{cc} \text{weights of subsets including 1} & \text{weights of subsets excluding 1} \end{array}}{\left( \begin{array}{cc} U_1 & U_3 \\ 0 & U_2 \end{array} \right)}$$

In doing so, we first observe that $U_1$ is exactly $M_{(n-1)(t-1)(k-1)}$ and is full rank from the induction hypothesis on $k$. Secondly, $U_2 = M_{(n-1)tk}$ and is thus full rank by the induction hypothesis on $n$. Therefore the matrix $M_{ntk}$ is full rank which concludes the inductive step on $n$. But this also concludes the inductive step on $k > 1$, and thus we we get that all $M_{ntk}$ along the antidiagonal of $M$ are full rank. It follows that $M$ is full rank, and thus $M\boldsymbol{w} = \boldsymbol{v}$ has a unique solution.

Finally, because we are choosing $k$ subset sizes from $[k, n-1]$, it's easy to see that if we sort subset size $s$ by $\binom{n}{s}$, then the $j$th subset size in this sorted order $s_i$ is such that $\binom{n}{s_j} \geq \binom{n}{k-j+1}$, which means the above condition applies. $\square$

**Theorem 2** (Necessity for Identification). *Suppose $\mathcal{H}$ includes all $k^{th}$ order CGAs and $v^*$ is a $k^{th}$ order CGA. If $\mathcal{D}_P$ contains performances of teams of only $m < k$ different sizes, then $\mathcal{D}_P$ does not always identify $v^*$.*

*Proof.* We will provide an instance when $k = 2$ such that $v^*$ is not identified. In that case $m = 1$, and we may pick teams of size $n - 1$. That gives us $\binom{n}{n-1} = n$ rows which is fewer than the number of columns $\binom{n}{2} + \binom{n}{1}$. Thus there will be more than one solution.

Moreover, the conditions specified in Theorem 1 are also tight in the sense that: if we allowed $m = k$ subset sizes, but over a wider interval, then $\mathcal{D}$ does not always identify $v^*$. To see this, consider $k = 2$ again. Widening the interval means the inclusion of either subset size $k - 1$ or $n$.

If we can pick $k - 1$, consider $m = 2$ subset sizes $k - 1$ and $n - 1$, which together gives $\binom{n}{1} + \binom{n}{n-1}$ rows, which is fewer than the number of columns $\binom{n}{2} + \binom{n}{1}$.

If we can pick $n$, consider $m = 2$ subset sizes $n - 1$ and $n$, which together gives $\binom{n}{n-1} + \binom{n}{n}$ rows, which is fewer than the number of columns $\binom{n}{2} + \binom{n}{1}$.

$\square$

## A.2 PAC Analysis

Another natural paradigm through which we may analyze sample complexity of learning a CGA is the PAC framework. Before we proceed, a word about why PAC bounds are not our main focus for sample complexity. One drawback of PAC bounds we considered is that it is only with high probability that *most* coalition values are well approximated. Therefore, it could still be that there is one $\hat{v}(S)$ that is arbitrarily off. Thus, the resultant estimated Shapley value will inherit this large bias. Since we hope to use the estimated Shapley Value for fair credit assignment in practice, we opt for what may be considered more "pessimistic", exact identification guarantees similar to those in [38].

Below, we provide two results based on PAC and PMAC notions of approximation. We prove the result assuming that we have correct CGA order specification. The result follows similarly when a higher order than that of the true CF is specified.

Consider a random sample $S$ of $m$ $(C, v(C))$ data points with $C$ uniformly sampled from $2^A$. There are at most $m$ distinct coalitional values in that sample. Call them $\boldsymbol{v}_{\widehat{S}}$. We will solve $M_{\widehat{S}}^{nk}\hat{\boldsymbol{\omega}} = \boldsymbol{v}_{\widehat{S}}$ where $M_{\widehat{S}}^{nk}$ denotes the matrix consisting of all rows corresponding to coalitions in $\widehat{S}$. This is feasible since there exist $\boldsymbol{\omega}$ s.t $M^{nk}\boldsymbol{\omega} = \boldsymbol{v}$. Note that this step assuming feasibility relies on the CGA model being of order $k$ or higher; if not, $M_{\widehat{S}}^{nk}\hat{\boldsymbol{\omega}} = \boldsymbol{v}_{\widehat{S}}$ may not be feasible.

In both parts of the proposition below, we will appeal to uniform convergence results to show that this construction yields a $\hat{\boldsymbol{\omega}}$ and the corresponding $\hat{\boldsymbol{v}}$ such that it approximates $\boldsymbol{v}$ with high probability. In all the sample complexity results that follow, let $c$ denote a generic constant.

**Proposition 1.** *Suppose $\boldsymbol{v}$ is a kth order CGA model with parameter vector $\omega$ of bounded $\ell_1$ norm. Then, with a set $\widehat{S}$ of $(C, v(C))$ data points of size $m \geq c\left(\frac{d_k + \log(1/\Delta)}{\delta^2}\right)$ uniformly sampled from $2^A$, we may compute $\hat{\boldsymbol{\omega}}$ and its corresponding $\hat{\boldsymbol{v}}$ as above such that, with probability at least $1 - \Delta$ over the samples $\widehat{S}$:*

$$\Pr_{C \sim 2^A}[\hat{v}(C) = v(C)] \geq 1 - \delta$$

*Proof.* The proof is motivated by the observation that $\boldsymbol{\omega}$ may be viewed as a linear classifier with dimension $d_k$. Indeed, if $\boldsymbol{\omega}$ is the true weight, then $M^{nk}\boldsymbol{\omega} = \boldsymbol{v}$, which is equivalent to:

$$[-M_C^{nk}, v(C)]^T[\boldsymbol{\omega}, 1] \geq 0 \text{ and } [M_C^{nk}, -v(C)]^T[\boldsymbol{\omega}, 1] \geq 0 \text{ for all C}$$

where $M_C^{nk}$ denotes the row of $M_{nk}$ corresponding to coalition $C$ and $[\boldsymbol{a}, \boldsymbol{b}], \boldsymbol{a} \in \mathbb{R}^n, \boldsymbol{b} \in \mathbb{R}^m$ denotes the $n + m$-dimensional vector obtained by concatenation of $\boldsymbol{a}$ and $\boldsymbol{b}$. Thus, if we define a classification task with $2 \cdot 2^N$ data points that have features $[-M_C^{nk}, v(C)], [M_C^{nk}, -v(C)]$ and labels 1 for all the points, we know there exists a classifier $f(\boldsymbol{x}) = \text{sign}([\boldsymbol{\omega}, 1]^T\boldsymbol{x})$ which achieves zero loss; here we take the sign of 0 to be 1.

Define data distribution $\mathcal{D}$ to be the uniform distribution over these $2 \cdot 2^N$ data points. A draw of size $m$ from $\mathcal{D}$ may be simulated by sampling coalitions from the uniform distribution over $2^A$ and then for each chosen coalition $C$, randomly choosing between $[-M_C^{nk}, v(C)]$ and $[M_C^{nk}, -v(C)]$ with equal probability.

Now, we are ready to prove that the $\hat{v}$ satisfies the statement in Proposition 1. To do this, we use the uniform convergence result below (Theorem 6.8 from [39]):

**Lemma 2.** *Let $\mathcal{H}$ be a hypothesis class for the classifier, and let $f$ be the true underlying classifier. If $\mathcal{H}$ has VC-dimension $d$, then with*

$$m \geq c\left(\frac{d + \log\left(\frac{1}{\Delta}\right)}{\delta^2}\right)$$

*i.i.d data points $\boldsymbol{x}_1, ..., \boldsymbol{x}_m \sim \mathcal{D}$,*

$$\delta \geq \left|\Pr_{\boldsymbol{x} \sim D}[h(\boldsymbol{x}) \neq f(\boldsymbol{x})] - \frac{1}{m}\sum_{i=1}^m \mathbb{1}_{h(\boldsymbol{x}^i) \neq f(\boldsymbol{x}^i)}\right|$$

*for all $h \in \mathcal{H}$ and with probability $1 - \Delta$ over the sampled data points.*

By construction, the classifier defined by $\hat{\boldsymbol{\omega}}$, $h(x) = \text{sign}([\hat{\boldsymbol{\omega}}, 1]^Tx))$, achieves zero empirical risk on $\hat{S}$ since $h(\boldsymbol{x}^i) = 1 = f(\boldsymbol{x}^i)$. So, we apply the uniform convergence result Lemma 2 with $\delta/2$ to get that with probability $1 - \Delta$ over the sampled data points $\boldsymbol{x}$ from $\mathcal{D}$:

$$\frac{\delta}{2} \geq \Pr_{\boldsymbol{x} \sim D}[h(\boldsymbol{x}) \neq f(\boldsymbol{x})]$$

$$= \Pr_{\boldsymbol{x} \sim D}[[\hat{\boldsymbol{\omega}}, 1]^T \boldsymbol{x} < 0)]$$

$$= \frac{1}{2} \Pr_{C \sim 2^A}[M_C^{nk} \hat{\boldsymbol{\omega}} > v_C] + \frac{1}{2} \Pr_{C \sim 2^A}[M_C^{nk} \hat{\boldsymbol{\omega}} < v_C]$$

$$= \frac{1}{2} \left( 1 - \Pr_{C \sim 2^A}[M_C^{nk} \hat{\boldsymbol{\omega}} = v_C] \right)$$

Therefore, the guarantee for $\hat{\boldsymbol{\omega}}$ over distribution $\mathcal{D}$ translates to the guarantee over the uniform distribution $2^A$ that $\hat{v} = M^{nk} \hat{\boldsymbol{\omega}}$ can overpredict or underpredict for at most $\delta$ percent of all coalitions.

To finish, we note that $[\boldsymbol{\omega}, 1]$ belongs to the hypothesis class of linear classifiers of dimension $d_k + 1$, which is known to have VC Dimension $d_k + 1$. So $\mathcal{H} = \{[\boldsymbol{\omega}, 1] \mid \boldsymbol{\omega} \in \mathbb{R}^{d_k}\}$ has VC dimension $d \leq d_k + 1$. And so, our sample complexity needed for $\hat{\boldsymbol{\omega}}$ to attain small generalization risk using Lemma 2 is $O(\frac{d_k + \log(1/\Delta)}{\delta^2})$.

$\square$

We remark that the sample complexity needed is on the same order as that shown by Theorem 1 in section A.1.

Next, we provide a PMAC-like guarantee [3] with much smaller sample complexity.

**Proposition 2.** *With samples of size $m \geq c \left( \frac{\log(d_k) + \log(1/\Delta)}{\epsilon^2 \delta^2} \right)$ uniformly sampled from $2^A$, we may compute $\hat{\boldsymbol{\omega}}$ and its corresponding $\hat{\boldsymbol{v}}$ as above such that, with probability at least $1 - \Delta$ over the samples:*

$$\Pr_{C \sim 2^A} [(1 - \epsilon)\hat{v}(C) \leq v(C) \leq (1 + \epsilon)\hat{v}(C)] \geq 1 - \delta$$

*Proof.* The proof follows from combining two known theorems adapted to our setting.

The left hand side of the probabilistic guarantee follows from a straightforward adaptation of the proof of Theorem 5 in [5]. In particular, the only tweak to the proof is that the features of the data points are to be instantiated as $M_C^{nk}/v(C)$ instead of $\mathbb{1}_C/v(C)$. Since $\boldsymbol{\omega}$ is bounded by our assumption, we do not need to bound it in terms of values of $v(C)$'s as is done in the proof of [5]. We note the loss function would then be defined as $\ell(\boldsymbol{\omega}, (M_C^{nk}/v(C), y)) = [\frac{M_C^{nk}\boldsymbol{\omega}}{v(C)} - 1]_+$ and $\hat{\boldsymbol{\omega}}$ achieves zero empirical loss because $M_{\widehat{S}}^{nk} \hat{\boldsymbol{\omega}} = \boldsymbol{v}_{\widehat{S}} \Rightarrow M_C^{nk} \hat{\boldsymbol{\omega}} = v_C \Rightarrow \frac{M_C^{nk}\hat{\boldsymbol{\omega}}}{v(C)} - 1 = 0$ for all $C \in S$. Altogether, we may arrive at the statement below:

*With a set of $m \geq c \left( \frac{\log(d_k) + \log(1/\Delta)}{\epsilon^2 \delta^2} \right)$ coalitions uniformly sampled from $2^A$, $\hat{\boldsymbol{\omega}}$ constructed as above is such that:*

$$\Pr_{C \sim 2^A} \left[ (1 - \epsilon) M_C^{nk} \hat{\boldsymbol{\omega}} \leq v(C) \right] \geq 1 - \delta$$

*with probability at least $1 - \Delta$ over the samples.*

The right hand side follows from a related theorem, Theorem 2 in [42] with the same change in the data features. Again, we can verify that $\hat{\boldsymbol{\omega}}$ achieves zero empirical loss:

*With a set of $m \geq c \left( \frac{\log(d_k) + \log(1/\Delta)}{\epsilon^2 \delta^2} \right)$ coalitions uniformly sampled from $2^A$, $\hat{\boldsymbol{\omega}}$ constructed as above is such that:*

$$\Pr_{C \sim 2^A}[v(C) \leq (1 + \epsilon) M_C^{nk} \hat{\boldsymbol{w}}] \geq 1 - \delta$$

*with probability at least $1 - \Delta$ over the samples.*

With this, we can initialize both theorems with $\Delta/2$ and $\delta/2$. We first union bound over the random draw of $m-$ size samples to conclude that with probability $\geq 1 - \Delta$, both inequalities hold for $\hat{\boldsymbol{\omega}}$, meaning that by union bound again for the random draw of $C$ over $2^A$:

$$\Pr_{C \sim 2^A}[(1 - \epsilon)M_C^{nk}\hat{\boldsymbol{w}} \leq v(C) \leq (1 + \epsilon)M_C^{nk}\hat{\boldsymbol{w}}] \geq 1 - \delta$$

$\square$

In summary, this means that under an even looser definition of approximability of the CGA model, the sample complexity needed is much smaller: only $O(\log(d_k))$ number of points are needed. Since $d_k \leq 2^n$, this means at most $O(n)$ samples are needed to estimate *most* of the coalition values *approximately* with high probability.

**Remark:** more generally, we may obtain the above two guarantees under the same sample complexity for any setting where we are looking to estimate solutions $\boldsymbol{x}$ to large scale linear programs $A\boldsymbol{x} = \boldsymbol{b}$, knowing apriori that $\|x\|_1$ is bounded. In such cases, we may obtain a PAC and PMAC-like result by computing $\hat{\boldsymbol{x}}$ from randomly sampled constraints $\boldsymbol{a}_i^T \boldsymbol{x} = b_i$. Notice here that $A \in \mathbb{R}^{2^n \times d_k}$ and the PMAC notion avoids needing the exponential sample complexity that is required to construct $\boldsymbol{b}$ to compute an exact solution.

This result may be of independent interest.

## A.3   Shapley Noise Bound Theorem Proofs

**Theorem 3** (Shapley noise L2 bound). *The L2 norm of the estimation error of the Shapley values is bounded by:*

$$\sum_{i=1}^{n} \left(\varphi_i(v) - \varphi_i(\hat{v})\right)^2 \leq \frac{2}{n} \sum_{C \in 2^A} \left(v(C) - \hat{v}(C)\right)^2 \tag{4}$$

*Proof.* First we observe that the Shapley value is a linear map $\mathbb{R}^{2^n} \to \mathbb{R}^n$ taking $v$ to $\varphi(v)$. We may describe this map with matrix $S_n \in \mathbb{R}^{n \times 2^n}$ where $n$ is the number of players in the cooperative game. Our work extends a line of work that studies properties of $S_n$, including [6] that studies its nullspace.

We have that:

$$||\varphi(v) - \varphi(\hat{v})||_2 = ||S_n v - S_n \hat{v}||_2 \leq ||S_n||_{op}||v - \hat{v}||_2$$

It suffices then to obtain the operator norm of $S_n$. We know that $||S_n||_{op} = \sqrt{\sigma_{\max}(S_n^T S_n)}$. $S_n^T S_n$ is complicated to analyze, so we opt to analyze $\sigma_{\max}(S_n S_n^T)$ since we know that the nonzero eigenvalues of $S_n^T S_n$ are the same as those of $S_n S_n^T$. $S_n S_n^T$ has nice structure in that all its off-diagonal entries are the same and all its diagonal entries are the same.

Take the $i$th row of $S_n$, $(S_n)_i$, we know that the entry in this row corresponding to subset $S$ is:

1. $\frac{1}{n}\binom{n-1}{|S|-1}^{-1}$ if $i \in S$

2. $-\frac{1}{n}\binom{n-1}{|S|}^{-1}$ if $i \notin S$

Therefore, let $d_1$ denote its diagonal entries, then:

$$d_1 = (S_n)_i^T (S_n)_i$$

$$= \sum_{S \in 2^{[n]}, i \in S} \left( \frac{1}{n} \binom{n-1}{|S|-1}^{-1} \right)^2 + \sum_{S \in 2^{[n]}, i \notin S} \left( -\frac{1}{n} \binom{n-1}{|S|}^{-1} \right)^2$$

$$= \frac{1}{n^2} \sum_{k=1}^{n} \binom{n-1}{k-1} \binom{n-1}{k-1}^{-2} + \frac{1}{n^2} \sum_{k=0}^{n-1} \binom{n-1}{k} \binom{n-1}{k}^{-2}$$

Let $d_2$ denote its off-diagonal entries. Consider the $i, j$th entry of $S_n S_n^T$, we can characterize the weights in the dot product as follows:

1. $\left( \frac{1}{n} \binom{n-1}{|S|-1}^{-1} \right)^2$ if $i, j \in S$

2. $\left( \frac{1}{n} \binom{n-1}{|S|-1}^{-1} \right) \left( -\frac{1}{n} \binom{n-1}{|S|}^{-1} \right)$ if $i \in S, j \notin S$

3. $\left( -\frac{1}{n} \binom{n-1}{|S|}^{-1} \right) \left( \frac{1}{n} \binom{n-1}{|S|-1}^{-1} \right)$ if $i \notin S, j \in S$

4. $\left( -\frac{1}{n} \binom{n-1}{|S|}^{-1} \right)^2$ if $i, j \notin S$

Therefore, when we sum these together:

$$d_2 = (S_n)_i^T (S_n)_j$$

$$= \sum_{S \in 2^{[n]}, i, j \in S} \left( \frac{1}{n} \binom{n-1}{|S|-1}^{-1} \right)^2 - \sum_{S \in 2^{[n]}, i \in S, j \notin S} \left( \frac{1}{n} \binom{n-1}{|S|-1}^{-1} \right) \left( -\frac{1}{n} \binom{n-1}{|S|}^{-1} \right)$$

$$- \sum_{S \in 2^{[n]}, i \notin S, j \in S} \left( -\frac{1}{n} \binom{n-1}{|S|}^{-1} \right) \left( \frac{1}{n} \binom{n-1}{|S|-1}^{-1} \right) + \sum_{S \in 2^{[n]}, i, j \notin S} \left( -\frac{1}{n} \binom{n-1}{|S|}^{-1} \right)^2$$

$$= \sum_{k=2}^{n} \binom{n-2}{k-2} \binom{n-1}{k-1}^{-2} - 2 \sum_{k=1}^{n-1} \binom{n-2}{k-1} \binom{n-1}{k}^{-1} \binom{n-1}{k-1}^{-1} + \sum_{k=0}^{n-2} \binom{n-2}{k} \binom{n-1}{k}^{-2}$$

It's easy to check that $d_1 > d_2$ since $d_2 = (S_n)_i^T (S_n)_j \leq ||(S_n)_i||_2 ||(S_n)_j||_2 = (S_n)_i^T (S_n)_i = d_1$.

And so, we may write:

$$S_n S_n^T = (d_1 - d_2) I_n + d_2 1_n$$

where $1_n$ is the all ones matrix.

This allows us to characterize all the eigenvalues of $S_n S_n^T$ and in particular the biggest one.

If the SVD of $1_n = U D U^T$, then we know that $D$ is a diagonal matrix with one entry being $n$ as this is an eigenvalue of $1_n$ and the rest being $0$ since $1_n$ is only rank 1. And so,

$$S_n S_n^T = U[(d_1 - d_2) I_n + d_2 D] U^T$$

This means that the top eigenvalue is $d_1 - d_2 + n \cdot d_2$ and the rest are all $d_1 - d_2$.

Evaluating $d_1 - d_2 + n \cdot d_2 = d_1 + (n-1) d_2$:

$$d_1 + (n-1)d_2 = \frac{1}{n^2}(\sum_{k=2}^{n-2} \binom{n-1}{k-1}^{-1} + \binom{n-1}{k}^{-1}$$

$$+ (n-1)[\frac{k-1}{n-1}\binom{n-1}{k-1}^{-1} - 2\frac{k}{n-1}\binom{n-1}{k-1}^{-1} + \binom{n-2}{k}\binom{n-1}{k}^{-2}]) + \frac{1}{n^2}r$$

$$= \frac{1}{n^2}((\sum_{k=2}^{n-2} k\binom{n-1}{k-1}^{-1} + \binom{n-1}{k}^{-1} - 2k\binom{n-1}{k-1}^{-1} + (n-1)\binom{n-2}{k}\binom{n-1}{k}^{-2}) + \frac{1}{n^2}r$$

$$= \frac{1}{n^2}((\sum_{k=2}^{n-2} -k\binom{n-1}{k-1}^{-1} + \binom{n-1}{k}^{-1} + (n-1-k)\binom{n-1}{k}^{-1}) + \frac{1}{n^2}r$$

$$= \frac{1}{n^2}((\sum_{k=2}^{n-2} -\frac{k!(n-k)!}{(n-1)!} + (n-k)\frac{k!(n-1-k)!}{(n-1)!}) + \frac{1}{n^2}r$$

$$= \frac{1}{n^2}r$$

It just remains to evaluate $r$ which are the residual terms from the sums, they are:

$$r = [1 + 1 + \frac{1}{n-1}] + [1 + \frac{1}{n-1} + 1] \text{ (from the two sums in } d_1)$$

$$+ (n-1)(([1 + \frac{n-2}{(n-1)^2}] - 2[\frac{1}{n-1} + \frac{1}{n-1}] + [1 + \frac{(n-2)}{(n-1)^2}]) \text{ (from the three sums in } d_2)$$

$$= 4 + \frac{2}{n-1} + 2n - 2 - 4 + \frac{2(n-2)}{n-1}$$

$$= 2n$$

To summarize, we get that:

$$\sigma_{\max}(S_n S_n^T) = d_1 + (n-1)d_2 = \frac{1}{n^2}2n = \frac{2}{n}$$

$$\Rightarrow ||S_n||_{op} = \sqrt{\sigma_{\max}(S_n^T S_n)} = \sqrt{\sigma_{\max}(S_n S_n^T)} = \sqrt{\frac{2}{n}}$$

which proves that $||\varphi(v) - \varphi(\hat{v})||_2 \le \sqrt{\frac{2}{n}}||v - \hat{v}||_2$ (4), as desired.

$\square$

The above is a worst case analysis by computing the largest singular value of the Shapley matrix. It turns out, most singular values of the Shapley matrix are very small and won't lead to a large amplification of the noise in the characteristic function.

We perform average case analysis by assuming that the error in the characteristic function is drawn uniformly from a smooth distribution, which is not very "peaky" anywhere, over all noise $v - \hat{v}$ with the same L2 norm.

**Theorem 4.** *Assuming that $v - \hat{v}$ is drawn from distribution $\mathcal{D}_{B_r}$ with support equal to a sphere and smooth in that $\kappa_0 \le \Pr_{\mathcal{D}_{B_r}}(x) \le \kappa_1$ for any point $x$ in its support, then:*

$$\mathbb{E}_{v-\hat{v} \sim \mathcal{D}_{B_r}}[||\varphi(v) - \varphi(\hat{v})||_2^2] \le \frac{6}{n}\frac{\kappa_1}{\kappa_0}\frac{||v - \hat{v}||_2^2}{2^n} \tag{5}$$

*Proof.* To obtain the bound in the theorem, we first prove the lemma below:

**Lemma 3.** *Let $\mathcal{D}_{B_r}$ be a distribution with support equal to a sphere with radius $r$ and smooth in that $\kappa_0 \leq \mathrm{Pr}_{\mathcal{D}_{B_r}}(x) \leq \kappa_1$ for any $x$ in its support. Consider any matrix $A \in \mathbb{R}^{m_1 \times m_2}$:*

$$\mathbb{E}_{x \sim \mathcal{D}_{B_r}}[\|Ax\|_2^2] \leq \frac{\kappa_1}{\kappa_0} \frac{\mathrm{Tr}(A^T A)}{m_2} \cdot r^2$$

*Proof.* Since $A^T A$ is symmetric and thus diagonalizable, consider its $m_2$ orthonormal eigenvectors $u_1, .., u_{m_2}$. We know that $u_1, .., u_{m_2}$ forms a basis of $\mathbb{R}^{m_2}$ and we can then write any $x$ in the support of $\mathcal{D}_{B_r}$ as $\sum_{j=1}^{m_2} \alpha_j u_j$. Moreover,

$$r^2 = \|x\|^2 = \left(\sum_{j=1}^{m_2} \alpha_j u_j\right)^T \left(\sum_{j=1}^{m_2} \alpha_j u_j\right) = \sum_{j=1}^{m_2} \alpha_j^2$$

since $u_j^T u_i = 0$ for $i \neq j$ and $\|u_j\|_2^2 = 1$.

Define $\mathcal{D}'_{B_r}$ to be the distribution over $\alpha$ that corresponds to each $x$ drawn from $\mathcal{D}_{B_r}$ and set $S_{D'}$ be its support (which may be characterized as a $m_2$ dimensional standard simplex as defined by $(\alpha_1^2/r^2, ..., \alpha_{m_2}^2/r^2)$). We abuse notation in letting $x(\alpha)$ be the corresponding $x$ to coefficients vector $\alpha$. It's a 1-1 correspondence, and so from the smoothness assumption on $\mathcal{D}_{B_r}$, $\mathrm{Pr}_{\mathcal{D}'_{B_r}}(\alpha) = \mathrm{Pr}_{\mathcal{D}_{B_r}}(x(\alpha)) \in [\kappa_0, \kappa_1]$.

Define $k^* = \arg\max_{k \in [m_2]} \mathbb{E}_{\alpha \sim \mathcal{D}'_{B_r}}[\alpha_k^2]$, then for any $i \neq k^*$:

$$
\begin{aligned}
\mathbb{E}_{\alpha \sim \mathcal{D}'_{B_r}}[\alpha_{k^*}^2] &= \int_{S_{D'}} \alpha_{k^*}^2 \, \mathrm{Pr}_{\mathcal{D}'_{B_r}}(\alpha) d\alpha \\
&\leq \int_{S_{D'}} \alpha_{k^*}^2 \kappa_1 d\alpha \\
&= \int_{S_{D'}} \alpha_i^2 \kappa_1 d\alpha \\
&\leq \int \alpha_i^2 \frac{\kappa_1}{\kappa_0} \mathrm{Pr}_{\mathcal{D}'_{B_r}}(\alpha) d\alpha \\
&= \frac{\kappa_1}{\kappa_0} \mathbb{E}_{\alpha \sim \mathcal{D}'_{B_r}}[\alpha_i^2]
\end{aligned}
$$

where the second equality follows from the symmetry of the support of $\mathcal{D}_{B_r}$, which is a sphere. This implies that:

$$\mathbb{E}_{\alpha \sim \mathcal{D}'_{B_r}}[\alpha_{k^*}^2] \leq \frac{\sum_{j=1}^{m_2} \frac{\kappa_1}{\kappa_0} \mathbb{E}_{\alpha \sim \mathcal{D}'_{B_r}}[\alpha_j^2]}{m_2} = \frac{\kappa_1}{\kappa_0} \frac{\mathbb{E}_{\alpha \sim \mathcal{D}'_{B_r}}[\sum_{j=1}^{m_2} \alpha_j^2]}{m_2} = \frac{\kappa_1}{\kappa_0} \frac{r^2}{m_2}$$

Therefore:

$$\mathbb{E}_{\boldsymbol{x} \sim \mathcal{D}_{B_r}}[\|A\boldsymbol{x}\|_2^2] = \mathbb{E}_{\boldsymbol{x} \sim \mathcal{D}_{B_r}}[\boldsymbol{x}^T A^T A \boldsymbol{x}]$$

$$= \mathbb{E}_{\boldsymbol{x} \sim \mathcal{D}_{B_r}}[\boldsymbol{x}^T (\sum_{j=1}^{m_2} \alpha_j \lambda_j \boldsymbol{u_j})]$$

$$= \mathbb{E}_{\boldsymbol{\alpha} \sim \mathcal{D}'_{B_r}}[\sum_{j=1}^{m_2} \lambda_j \alpha_j^2]$$

$$= \sum_{j=1}^{m_2} \lambda_j \mathbb{E}_{\boldsymbol{\alpha} \sim \mathcal{D}'_{B_r}}[\alpha_j^2]$$

$$\leq \mathbb{E}_{\boldsymbol{\alpha} \sim \mathcal{D}'_{B_r}}[\alpha_{k^*}^2] \sum_{j=1}^{m_2} \lambda_j$$

$$\leq \frac{\kappa_1}{\kappa_0} \frac{r^2}{m_2} \sum_{j=1}^{m_2} \lambda_j$$

$\square$

Using this Lemma 3, we can then perform an average case analysis:

$$\mathbb{E}[\|S_n \boldsymbol{x}\|_2^2] \leq \frac{\kappa_1}{\kappa_0} \frac{\mathrm{Tr}(S_n^T S_n)}{2^n} \|\boldsymbol{x}\|_2^2 = \frac{\kappa_1}{\kappa_0} \frac{\mathrm{Tr}(S_n S_n^T)}{2^n} \|\boldsymbol{x}\|_2^2$$

We know that $\mathrm{Tr}(S_n S_n^T) = nd_1$ so the average case multiplier of the noise is:

$$d_1 = \frac{1}{n^2} \sum_{k=1}^{n} \binom{n-1}{k-1}^{-1} + \frac{1}{n^2} \sum_{k=0}^{n-1} \binom{n-1}{k}^{-1}$$

$$= \frac{1}{n^2}(2 + \sum_{k=1}^{n-1} \binom{n-1}{k-1}^{-1} + \binom{n-1}{k}^{-1})$$

$$= \frac{2}{n^2} + \frac{1}{n^2}(\sum_{k=1}^{n-1} \frac{(k-1)!(n-k-1)!(k+n-k)}{(n-1)!})$$

$$= \frac{2}{n^2} + \frac{1}{n(n-1)}(\sum_{k=1}^{n-1} \binom{n-2}{k-1}^{-1})$$

$$= \frac{2}{n^2} + \frac{2}{n(n-1)} + \frac{1}{n(n-1)}(\sum_{k=2}^{n-2} \binom{n-2}{k-1}^{-1})$$

$$\leq \frac{2}{n^2} + \frac{2}{n(n-1)} + \frac{1}{n(n-1)}((n-3)\binom{n-2}{1}^{-1})$$

$$= \frac{2}{n^2} + \frac{3n-7}{n(n-1)(n-2)}$$

$$\leq \frac{6}{n^2}$$

So, the multiplier is $\frac{\kappa_1}{\kappa_0} \frac{6}{n 2^n}$ over the distribution $\mathcal{D}_{B_r}$. $\square$

Next, we can obtain a more general result by integrating across all L2 norms $r$ that $v - \hat{v}$ can take.

**Corollary 1.** *Suppose noise* $v - \hat{v} \sim \mathcal{D}_n$ *is such that its conditional distribution satisfies* $\kappa_0(r) \leq \Pr_{\mathcal{D}_n}(x | \|x\|_2^2 = r^2) \leq \kappa_1(r)$ *for all* $r$ *and* $x$ *in* $\mathcal{D}_n$*'s support, then:*

$$\mathbb{E}_{v-\hat{v} \sim \mathcal{D}_n}[\|\varphi(v) - \varphi(\hat{v})\|_2^2] \leq \frac{6}{n} \mathbb{E}_r \left[ \frac{\kappa_1(r)}{\kappa_0(r)} \left( \frac{r^2}{2^n} \right) \right]$$

*Proof.* This follows from iterated expectation:

$$\mathbb{E}_{v-\hat{v} \sim \mathcal{D}}[\|\varphi(v) - \varphi(\hat{v})\|_2^2] = \mathbb{E}_r[\mathbb{E}_{v-\hat{v} \sim \mathcal{D}_{B_r}}[\|\varphi(v) - \varphi(\hat{v})\|_2^2 \mid \|v - \hat{v}\|_2^2 = r^2]]$$

$$\leq \mathbb{E}_r \left[ \left( \frac{\kappa_1(r)}{\kappa_0(r)} \frac{6}{n2^n} \right) r^2 \right]$$

$$= \frac{6}{n2^n} \mathbb{E}_r \left[ \frac{\kappa_1(r)}{\kappa_0(r)} r^2 \right]$$

where the inequality holds by Theorem 4.

$\square$

**Remark:** Therefore, if $\mathbb{E}_r[\frac{\kappa_1(r)}{\kappa_0(r)} r^2] = c\mathbb{E}_r[r^2]$ for some constant $c = O(1)$, then the error in the Shapley value is fairly small and proportional to $O(1/n)$ of the average L2 error of $v - \hat{v}$.

**Theorem 5** (Shapley noise L1 bound). *The sum of absolute errors in Shapley values is bounded by:*

$$\sum_{i=1}^n |\varphi_i(v) - \varphi_i(\hat{v})| \leq \sum_{C \in 2^A} |v(C) - \hat{v}(C)| \tag{6}$$

*Assuming there is no error in estimating the grand coalition nor the empty set and* $n \geq 3$*, then we can give a stronger bound on the sum of absolute errors:*

$$\sum_{i=1}^n |\varphi_i(v) - \varphi_i(\hat{v})| \leq \frac{2}{n} \sum_{C \in 2^A} |v(C) - \hat{v}(C)| \tag{7}$$

*Furthermore, assume players are divided into m equal sized teams,* $G_1, ..., G_m$*, where* $|G_i| = N/m$*. Then if we compute their Shapley values just with respect to their own teams we get:*

$$\sum_{i=1}^n |\varphi_i(v) - \varphi_i(\hat{v})| \leq \frac{2m}{n} \sum_{C \in 2^A} |v(C) - \hat{v}(C)| \tag{8}$$

*Proof.* We can express the difference in Shapley value for $i$ as:

$$|\varphi_i(v) - \varphi_i(\hat{v})| = |\frac{1}{n} \sum_{S \subseteq [n] \setminus \{i\}} \binom{n-1}{|S|}^{-1} ([v(S \cup \{i\}) - \hat{v}(S \cup \{i\})] - [v(S) - \hat{v}(S)])|$$

$$\leq \frac{1}{n} \sum_{S \subseteq [n] \setminus \{i\}} \binom{n-1}{|S|}^{-1} (|v(S \cup \{i\}) - \hat{v}(S \cup \{i\})| + |v(S) - \hat{v}(S)|)$$

$$= \frac{1}{n} \sum_{s=0}^{n-1} \binom{n-1}{s}^{-1} \sum_{S \subseteq [n] \setminus \{i\}, |S|=s} (|v(S \cup \{i\}) - \hat{v}(S \cup \{i\})| + |v(S) - \hat{v}(S)|)$$

Thus, for any $S$ of size $s$:

- If it contains element $i$, its L1 $v$ error is weighted by $\binom{n-1}{s-1}^{-1}$.

- If it doesn't, it is weighted by $\binom{n-1}{s}^{-1}$.

Observe that the unweighted RHS is equal to:

$$= \frac{1}{n} \sum_{s=0}^{n-1} \sum_{S \subseteq [n] \setminus \{i\}, |S|=s} \left( |v(S \cup \{i\}) - \hat{v}(S \cup \{i\})| + |v(S) - \hat{v}(S)| \right)$$

$$= \frac{1}{n} \sum_{S \subseteq [n] \setminus \{i\}} |v(S \cup \{i\}) - \hat{v}(S \cup \{i\})| + \sum_{S \subseteq [n] \setminus \{i\}} |v(S) - \hat{v}(S)|$$

$$= \frac{1}{n} ||v - \hat{v}||_1$$

Therefore, since $\binom{n-1}{s}^{-1} \le 1$ for $s \in [0, n-1]$:

$$|\varphi_i(v) - \varphi_i(\hat{v})| \le \frac{1}{n} \sum_{s=0}^{n-1} \binom{n-1}{s}^{-1} \sum_{S \subseteq [n] \setminus \{i\}, |S|=s} \left( |v(S \cup \{i\}) - \hat{v}(S \cup \{i\})| + |v(S) - \hat{v}(S)| \right)$$

$$\le \frac{1}{n} ||v - \hat{v}||_1$$

Summing across all $i$'s, this proves inequality (6).

Note that $\binom{n-1}{s}^{-1} = 1$ holds only for (i) the full set $[n]$ ($s = n - 1$) (ii) the set $\{i\}$ ($s = 0$) (iii) empty set ($s = 0$) (iv) set $[n] \setminus \{i\} = [-i]$ ($s = n - 1$). Thus we can obtain equality if all of the errors in $v$ lie in estimating the full set or the empty set. This makes the bound tight.

We obtain a stronger inequality (7) if we assume that there is no error in estimating the empty nor the grand coalition value:

Let $e = ||v - \hat{v}||_1$ and $e_i = |v(\{i\}) - \hat{v}(\{i\})| + |v([-i]) - \hat{v}([-i])|$. Then:

$$|\varphi_i(v) - \varphi_i(\hat{v})| \le \frac{1}{n} \sum_{s=0}^{n-1} \binom{n-1}{s}^{-1} \sum_{S \subseteq [n] \setminus \{i\}, |S|=s} \left( |v(S \cup \{i\}) - \hat{v}(S \cup \{i\})| + |v(S) - \hat{v}(S)| \right)$$

$$\le \frac{1}{n} e_i + \frac{1}{n} \sum_{s=1}^{n-2} \binom{n-1}{s}^{-1} \sum_{S \subseteq [n] \setminus \{i\}, |S|=s} \left( |v(S \cup \{i\}) - \hat{v}(S \cup \{i\})| + |v(S) - \hat{v}(S)| \right)$$

$$\le \frac{1}{n} e_i + \frac{1}{n(n-1)} (e - e_i)$$

since $\binom{n-1}{s}^{-1} \le \frac{1}{n-1}$ for $s \in [1, n-2]$.

Summing this across i gives:

$$|\varphi(v) - \varphi(\hat{v})| \le \frac{1}{n} \sum_{i=1}^{n} e_i + \frac{1}{n(n-1)} \left( ne - \sum_{i=1}^{n} e_i \right)$$

$$= \frac{e}{n-1} + \frac{n-2}{n(n-1)} \left( \sum_{i=1}^{n} e_i \right)$$

$$\le \frac{e}{n-1} + \frac{n-2}{n(n-1)} e$$

$$= \frac{2e}{n}$$

since $\sum_{i=1}^{n} e_i \leq e$.

This proves inequality (7).

In some case, as is the case with our NBA experimental setup, players are divided into $m$ groups, $G_1, ..., G_m$, and we wish to compute their Shapley values only with respect to their own groups. We can follow a similar analysis as above to derive a bound on the Shapley values. For player $i \in G_j$:

$$
\begin{aligned}
|\varphi_i(v) - \varphi_i(\hat{v})| &= |\frac{1}{|G_j|} \sum_{S \subseteq G_j \setminus \{i\}} \binom{|G_j| - 1}{|S|}^{-1} ([v(S \cup \{i\}) - \hat{v}(S \cup \{i\})] - [v(S) - \hat{v}(S)])| \\
&\leq \frac{1}{|G_j|} \sum_{S \subseteq G_j \setminus \{i\}} \binom{|G_j| - 1}{|S|}^{-1} (|v(S \cup \{i\}) - \hat{v}(S \cup \{i\})| + |v(S) - \hat{v}(S)|) \\
&= \frac{1}{|G_j|} \sum_{s=0}^{|G_j|-1} \binom{|G_j| - 1}{s}^{-1} \sum_{S \subseteq G_j \setminus \{i\}, |S|=s} (|v(S \cup \{i\}) - \hat{v}(S \cup \{i\})| + |v(S) - \hat{v}(S)|)
\end{aligned}
$$

Let $E_j = \sum_{S \subseteq G_j} |v(S) - \hat{v}(S)|$. It's clear that $\sum_{j=1}^{m} E_j \leq e$ since any two teams $G_{j_1}, G_{j_2}$ are disjoint for $j_1 \neq j_2$ and thus don't have any subsets in common; note our assumption that the empty set is estimated without any error by $\hat{v}$.

To maximize the cumulative error, all the errors in $e$ should be placed in subsets $S$ with $S \subseteq G_j$ for some $j$. So WLOG we can assume that $\sum_{j=1}^{m} E_j = e$. Using inequality (7), we get that:

$$
\sum_{i \in G_j} |\varphi_i(v) - \varphi_i(\hat{v})| \leq \frac{2E_j}{|G_j|}
$$

So the overall bound is:

$$
|\varphi(v) - \varphi(\hat{v})| \leq \sum_{j=1}^{m} \frac{2E_j}{|G_j|}
$$

Assume $|G_j| = n/m$ for each j, this simplifies to $\frac{2m}{n} e$ and proves inequality 8.

$\square$

While the above bounds are tight, the analysis is worst case. For instance, for the first bound we provide, equality holds when all the error in the $\|v - \hat{v}\|_1$ vector is in the coalition value of the grand coalition or the empty set. Below, we provide a simple, average case analysis to show that on average, a randomly drawn error vector leads to a small increase in L1 Shapley error in expectation.

**Theorem 6** (Average case Shapley noise L1 bound). *Assuming that the error $\boldsymbol{v} - \hat{\boldsymbol{v}}$ is such that the vector $|\boldsymbol{v} - \hat{\boldsymbol{v}}|/r$ (where absolute value is coordinate wise) is drawn from distribution $\mathcal{D}_{S_r}$ with support equal to the surface of a $2^n$-simplex and smooth in that $\kappa_0 \leq \mathrm{Pr}_{\mathcal{D}_{S_r}}(\boldsymbol{x}) \leq \kappa_1$ for any point $\boldsymbol{x}$ in its support, then:*

$$
\mathbb{E}_{\boldsymbol{v} - \hat{\boldsymbol{v}} \sim \mathcal{D}_{S_r}}[\|\varphi(\boldsymbol{v}) - \varphi(\hat{\boldsymbol{v}})\|_1] \leq 2 \frac{\kappa_1}{\kappa_0} \frac{\|\boldsymbol{v} - \hat{\boldsymbol{v}}\|_1}{2^n} \tag{9}
$$

*Proof.*

$$\mathbb{E}_{\boldsymbol{v}-\hat{\boldsymbol{v}}\sim\mathcal{D}_{S_r}}[|\varphi_i(\boldsymbol{v})-\varphi_i(\hat{\boldsymbol{v}})|] = \mathbb{E}_{\boldsymbol{v}-\hat{\boldsymbol{v}}\sim\mathcal{D}_{S_r}}\left[|\frac{1}{n}\sum_{S\subseteq[n]\backslash\{i\}}\binom{n-1}{|S|}^{-1}([v(S\cup\{i\})-\hat{v}(S\cup\{i\})]-[v(S)-\hat{v}(S)])|\right]$$

$$\leq \frac{1}{n}\sum_{s=0}^{n-1}\binom{n-1}{s}^{-1}\sum_{S\subseteq[n]\backslash\{i\},|S|=s}\Big(\mathbb{E}_{\boldsymbol{v}-\hat{\boldsymbol{v}}\sim\mathcal{D}_{S_r}}\big[||v(S\cup\{i\})-\hat{v}(S\cup\{i\})|\big]+$$

$$\mathbb{E}_{\boldsymbol{v}-\hat{\boldsymbol{v}}\sim\mathcal{D}_{S_r}}\big[|v(S)-\hat{v}(S)|\big]\Big)$$

$$\overset{(1)}{\leq} \frac{1}{n}\sum_{s=0}^{n-1}\binom{n-1}{s}^{-1}\sum_{S\subseteq[n]\backslash\{i\},|S|=s}\left(\frac{\kappa_1}{\kappa_0}\frac{r}{2^n}+\frac{\kappa_1}{\kappa_0}\frac{r}{2^n}\right)$$

$$= \frac{2}{n}\frac{\kappa_1}{\kappa_0}\frac{r}{2^n}$$

where $(1)$ is due to the following:

Let subset $C^* = \arg\max_C \mathbb{E}_{\boldsymbol{v}-\hat{\boldsymbol{v}}\sim\mathcal{D}_{S_r}}[||v(C)-\hat{v}(C)|]$ and subset $C' = \arg\min_C \mathbb{E}_{\boldsymbol{v}-\hat{\boldsymbol{v}}\sim\mathcal{D}_{S_r}}[||v(C)-\hat{v}(C)|]$:

$$\mathbb{E}_{\boldsymbol{v}-\hat{\boldsymbol{v}}\sim\mathcal{D}_{S_r}}[||v(C^*)-\hat{v}(C^*)|] = \int |v(C^*)-\hat{v}(C^*)|\operatorname{Pr}_{\mathcal{D}_{S_r}}(\boldsymbol{v}-\hat{\boldsymbol{v}})d(\boldsymbol{v}-\hat{\boldsymbol{v}})$$

$$\leq \int |v(C^*)-\hat{v}(C^*)|\kappa_1 d(\boldsymbol{v}-\hat{\boldsymbol{v}})$$

$$\overset{(2)}{=} \int |v(C')-\hat{v}(C')|\kappa_1 d(\boldsymbol{v}-\hat{\boldsymbol{v}})$$

$$\leq \int |v(C')-\hat{v}(C')|\frac{\kappa_1}{\kappa_0}\operatorname{Pr}_{\mathcal{D}_{S_r}}(\boldsymbol{v}-\hat{\boldsymbol{v}})d(\boldsymbol{v}-\hat{\boldsymbol{v}})$$

$$= \frac{\kappa_1}{\kappa_0}\mathbb{E}_{\boldsymbol{v}-\hat{\boldsymbol{v}}\sim\mathcal{D}_{S_r}}[||v(C')-\hat{v}(C')|]$$

$$\overset{(3)}{\leq} \frac{\kappa_1}{\kappa_0}\left(\frac{r}{2^n}\right).$$

Here $(2)$ holds by symmetry as the expectation of any two vector coordinates under a uniform distribution over the simplex of vectors is the same. $(3)$ holds because every vector in the support of $\mathcal{D}_{S_r}$ has L1 norm of $r$, $\sum_C \mathbb{E}_{\boldsymbol{v}-\hat{\boldsymbol{v}}\sim\mathcal{D}_{S_r}}[|v(C)-\hat{v}(C)|] = \mathbb{E}_{\boldsymbol{v}-\hat{\boldsymbol{v}}\sim\mathcal{D}_{S_r}}[\sum_C |v(C)-\hat{v}(C)|] = r$ and so by our choice of $C'$, $\mathbb{E}_{\boldsymbol{v}-\hat{\boldsymbol{v}}\sim\mathcal{D}_{S_r}}[||v(C')-\hat{v}(C')|] \leq \frac{r}{2^n}$.

Summing $\mathbb{E}_{\boldsymbol{v}-\hat{\boldsymbol{v}}\sim\mathcal{D}_{S_r}}[|\varphi_i(\boldsymbol{v})-\varphi_i(\hat{\boldsymbol{v}})|]$ across all $i$ gives the result.

$\square$

This means that, on average, for a randomly drawn $\varphi(\boldsymbol{v})-\varphi(\hat{\boldsymbol{v}})$ with a fixed error budget in L1 error, the L1 error in the Shapley is only proportional to the average error in estimating each coalition. Next, we can obtain a more general bound by integrating across all L1 norms $r$ that $\varphi(\boldsymbol{v})-\varphi(\hat{\boldsymbol{v}})$ can take.

**Corollary 2.** *Suppose noise $v-\hat{v}\sim\mathcal{D}_n$ is such that its conditional distribution satisfies $\kappa_0(r) \leq \operatorname{Pr}_{\mathcal{D}_n}(x|\|x\|_1 = r) \leq \kappa_1(r)$ for all $r$ and $x$ in $\mathcal{D}_n$'s support, then:*

$$\mathbb{E}_{v-\hat{v}\sim\mathcal{D}_n}[\|\varphi(\boldsymbol{v})-\varphi(\hat{\boldsymbol{v}})\|_1] \leq 2\mathbb{E}_r\left[\frac{\kappa_1(r)}{\kappa_0(r)}\left(\frac{r}{2^n}\right)\right]$$

*Proof.* This follows from iterated expectation:

$$\mathbb{E}_{\boldsymbol{v}-\hat{\boldsymbol{v}}\sim\mathcal{D}}[\|\varphi(\boldsymbol{v}) - \varphi(\hat{\boldsymbol{v}})\|_1] = \mathbb{E}_r\left[\mathbb{E}_{\boldsymbol{v}-\hat{\boldsymbol{v}}\sim\mathcal{D}_{S_r}}\left[\|\varphi(\boldsymbol{v}) - \varphi(\hat{\boldsymbol{v}})\|_1 \mid \|\boldsymbol{v} - \hat{\boldsymbol{v}}\|_1 = r\right]\right]$$
$$\leq \mathbb{E}_r\left[\left(\frac{\kappa_1(r)}{\kappa_0(r)}\frac{2}{2^n}\right)r\right]$$
$$= 2\mathbb{E}_r\left[\frac{\kappa_1(r)}{\kappa_0(r)}\frac{r}{2^n}\right]$$

where the inequality holds by the Theorem above.

$\square$

**Remark:** Therefore, if $\mathbb{E}_r[\frac{\kappa_1(r)}{\kappa_0(r)}r] = c\mathbb{E}_r[r]$ for some constant $c = O(1)$, then the error in the Shapley value is fairly small and proportional to the average expected L1 error $\frac{\mathbb{E}_r[r]}{2^n}$.

### A.4 Discussion about CGA-Specific Errors:

Since CGA is a *complete representation*, every game may be expressed as a CGA of some order (see Fact 1). And so, we may plug the CGA-specific bias into the general bounds obtained previously in Theorems 3-6.

Below we derive CGA bias due to model misspecification. Note that the approximation is lossy only when the true game is generated by a CGA model of order $r$ and we model it with a simpler CGA model of order $k$ with $k < r$. When we model the game with a CGA of a higher order than the actual game, it is clear that we can learn a set of weights that would fit the coalition values exactly (since $v$ would be in the columnspace).

Let $M^{nk}$ denote the matrix relating the parameters $\boldsymbol{\omega}$ to the coalitional values $\boldsymbol{v}$. It is a $2^n \times d_k$ matrix of the form:

|  | first order weights | ... | $k$th order weights |
| --- | --- | --- | --- |
| row corresponding to null coalition {} | ... | ... | ... |
| ... | ... | ... | ... |
| row corresponding to grand coalition $A$ | ... | ... | ... |

The CGA model parameters $\hat{\boldsymbol{\omega}}$ we learn will be such that:

$$\hat{\boldsymbol{\omega}} = \arg\min_{\boldsymbol{w}} \|M^{nk}\boldsymbol{w} - M^{nr}\boldsymbol{\omega_r^*}\|_2^2$$

This is just equivalent to projecting vector $M^{nr}\boldsymbol{\omega_r^*}$ onto the columnspace spanned by $M^{nk}$. Recall from our identification theorem that $M^{nk}$ has enough rows to be full rank, which makes $(M^{nk})^T M^{nk}$ positive definite and invertible; if there are not enough samples, we may instead consider a regularization term that will make the matrix invertible. Define projection matrix $P_{nk}$:

$$P_{nk} = M^{nk}(((M^{nk})^T M^{nk})^{-1}(M^{nk})^T$$

This means that the misspecification error $e(n, k, r)$ may be expressed as:

$$e(n, k, r) = (I - P_{nk})M^{nr}\boldsymbol{\omega_r^*}$$

which we may plug into our noise bounds for the Shapley value computation.

Unlike the Shapley matrix, the error matrix $(I - P_{nk})M^{nr}$ does not seem to admit a closed form for its trace. Instead, we perform simulations to better understand its properties. In particular, we look to understand if it enjoys the same "averaging-effect" as the Shapley matrix. We compute the max eigenvalue and the average trace norm value sweeping over all $n, r, k$ for $k < r < n$ for $n \in [2, 15]$ (we try these sizes since 15 is the largest possible before the error matrix's size exceeds

that permitted by our machine memory). Our simulations suggest that its largest eigenvalue (for the worst case bound) and the average trace value (for the average case bound per Lemma 3) both grow monotonically with $n$ and $r$ (fixing a $k$). Altogether, this suggests that the $\ell_2$ error can grow arbitrarily large with model misspecification.

## A.5 For Practitioners: How to choose the order of the CGA Model

The order $k$ of the CGA model is dependent on the application at hand. It may be set to be the maximum $k$-way interaction that the practitioner expects to take place in the team.

Our model is especially useful in settings like the NBA, in which team sizes are small relative to the overall number of players. This structural prior can be encoded in the order of the CGA. As an example, for the NBA, we expect at most 5-way interaction and so it is necessary to only consider compact, low-rank models.

Note that for the time and space complexity of the model, the time to compute the SV is the space complexity of the CGA model: the number of parameters. The complexity of learning a CGA depends on the training method employed to learn $\hat{v}$ (e.g. we use SGD).

## A.6 Proofs of Facts

For completeness, we provide proofs of the two facts listed.

**Fact 1** (Unique decomposition form). *There exists a unique set of values $\omega_S$ for each subset $S \subseteq A$ with $|S| \leq k$ such that the characteristic function can be decomposed into its interaction form where*

$$v(C) = \sum_{k=1}^{|C|} \sum_{S \in 2_k^C} \omega_S. \tag{10}$$

*Proof.* We can show this inductively. For the base case when $|C| = 1$ we have $w_C = v(C)$, which is unique.

Induction step: assume $w_{S'}$ is uniquely determined for $|S'| = 1, ..., m - 1$. Then for a particular subset $|S| = m$:

$$v(S) = \sum_{i=1}^{m-1} \sum_{S' \in 2_i^S} w_{S'} + w_S$$

and thus $w_S$ is uniquely determined since we must set it to

$$w(S) = v(S) - \sum_{i=1}^{m-1} \sum_{S' \in 2_i^S} w_{S'}$$

$\square$

**Fact 2** (Shapley value expression). *The Shapley Value of an individual $i$ with respect to team $A$ can be expressed as:*

$$\varphi_i(v) = \sum_{T \subseteq A \setminus \{i\}} \frac{1}{|T| + 1} \omega_{T \cup \{i\}}$$

*Proof.* The Shapley value for player $i$ is defined as:

$$\varphi_i(v) = \frac{1}{n} \sum_{S \subseteq A \setminus \{i\}} \binom{n-1}{|S|}^{-1} (v(S \cup \{i\}) - v(S))$$

Plugging in the decomposition form:

$$v(S \cup \{i\}) - v(S) = \sum_{S' \subseteq S} w_{S' \cup \{i\}}$$

Thus, the Shapley value for $i$ is only a function of all $w_S$ where $i \in S$.

Given a subset $T = \{i_1...i_t\}$, let us derive the weighted sum of $w_T$ occurrences in $\varphi_{i_1}(v)$. This term only appears if $\{i_2,..,i_t\} \subseteq S$ but $i_1 \notin S$. And so, the weighted sum of occurrences is:

$$\frac{1}{n} \sum_{S \subseteq A \setminus \{i_1\}, \{i_2,..,i_t\} \in S} \binom{n-1}{|S|}^{-1} = \frac{1}{n} \sum_{s=t-1}^{n-1} \binom{n-1}{s}^{-1} \binom{n-t}{s-(t-1)}$$

Similarly, $w_T$ has the same sum of weighted occurrences in expressions for players $i_2, ..., i_t$. And so, by efficiency (since $v(A)$ contains exactly $w_T$ and the sum of Shapley payments equals $v(A)$), they must be assigned equal portions of $w_T$, i.e $w_T/|T|$. This holds for all subsets T. And so, a player $i$'s Shapley value is the sum of all weights $w_T/|T|$, for all subsets $T \subseteq [n]$ and $i \in T$. $\qquad \square$

## A.7 Relationship to the Core

The main text of the paper has focused on the solution concept of the Shapley Value. Another commonly used solution concept in cooperative game theory is known as the Core [17]. Let $n$ be the number of players in the game, the Core is an allocation $x \in \mathbb{R}^n$ that satisfies:

(i) Efficiency: $\sum_{i \in [n]} x_i = v([n])$

(ii) Stability: for any coalition $C \subset [n]$:

$$\sum_{i \in C} x_i \geq v(C)$$

Intuitively, a payoff vector is in the Core if it incentivizes every coalition $C$ to stay with the grand coalition rather than leave, achieve a value of $v(C)$ and split it amongst themselves in some other way.

The Core of a game may be empty, though an extension known as the Least Core is always guaranteed to exist. The Least Core can be computed by solving the following linear program:

$$\begin{aligned} \min_{e,x} \quad & e \\ s.t. \quad & \sum_{i \in [n]} x_i = v([n]) \\ & \sum_{i \in C} x_i \geq v(C) - e \quad \forall S \subset [n] \end{aligned}$$

Intuitively, the Least Core is the allocation which minimizes the subsidy $e$ required to incentivize all coalitions to stay together. We call the minimum subsidy needed the Least Core value. Unfortunately, [13] show that for any CGA model with order higher than 1, it is NP-Complete to compute the Least Core Value.

One notable allocation in the Least Core is the Nucleolus. For a given allocation $x$, define deficit function $e_x(C) = v(C) - \sum_{i \in C} x_i$. Order all subsets of $[n]$ according to the deficit function $e_x$. The nucleolus is defined as the imputation which lexicographically minimizes this ordering of deficits. Intuitively, the Nucleolus is the "inner-most" allocation in the Least Core. In general, the Nucleolus is difficult to compute and requires solving a series of exponential-size linear programs.

Remarkably, [13] prove the following fact:

**Fact 3.** *Assuming the characteristic function of the underlying game is a second order CGA model, the Shapley Value is in the Least Core (in fact, it is the Nucleoulus).*

Therefore, we can simply compute the Shapley value to obtain a point in the Least Core. All that remains is to approximate the Least Core value. To do this, we establish an approximate notion of the Least Core value by adapting a similar notion from [5] and derive a simple sample complexity bound for estimating this value. The definition goes as follows:

**Definition 5.** *Given an allocation $x$, a value $e$ is a $\delta-$probable least core value if:*

$$\Pr_{C\sim 2^A}[\sum_{i\in C} x_i + e \geq v(C)] \geq 1 - \delta$$

The least core value is the smallest $e^*$ such that there exist an allocation for which $e^*$ is a $0-$probable least core value.

We will compute a $\delta-$probable least core value by computing the sample least core value on a set of uniformly sampled coalitions $\widehat{S}$. Certainly if $|\widehat{S}| = 2^n$ coalitions, then the sample least core value will be the true least core value exactly. Using standard learning theory tools, we can relate the quality of the estimation of the least core value, in terms of $\delta$, to the size of the samples $\widehat{S}$ needed:

**Theorem 7.** *Given a set $\widehat{S}$ of $m = O(\frac{\log(1/\Delta)}{\delta^2})$ coalitions uniformly sampled from $2^A$, let:*

$$\hat{e} = \arg\min e$$
$$\sum_{i\in C} \varphi_i(\hat{v}) \geq v(C) - e \quad \forall C \subseteq \widehat{S}$$

*then with probability $1 - \Delta$ over the samples, $\hat{e}$ is a $\delta-$least core value.*

*Proof.* We prove this through a simple learning theory setup analogous to the proposition above. Define a 2-dimensional linear classifier with weights $w_e = [e, 1]$. This class of classifier is a subset of all linear classifiers of dimension 2 and thus has VC dimension $\leq 2$.

For each of the $2^n - 2$ inequality constraints, construct data point $[1, \sum_{i\in C} \varphi_i(\hat{v}) - v(C)]$ that corresponds to coalition $C$'s constraint. We assign each data point a label of 1. Notice that if classifier $w_e = [e, 1]$ classifies $[1, \sum_{i\in C} \varphi_i(\hat{v}) - v(C)]$ correctly, then:

$$\text{sign}_{w_e}([1, \sum_{i\in C} \varphi_i(\hat{v}) - v(C)]) = 1 \Rightarrow [e, 1]^T[1, \sum_{i\in C} \varphi_i(\hat{v}) - v(C)] \geq 0 \Rightarrow \sum_{i\in C} \varphi_i(\hat{v}) \geq v(C) - e$$

Moreover, we know that the classifier we obtain, $w_{\hat{e}} = [\hat{e}, 1]$, is such that it classifies all the samples in $\widehat{S}$ correctly by construction, and has zero empirical risk. Again, using Lemma 2, we know that this classifier's performance on the samples generalize to all $2^n - 2$ constraints. In particular, if there are at least

$$O(\frac{2 + \log(1/\Delta)}{\delta^2})$$

samples in $\widehat{S}$, then the empirical least core value $\hat{e}$ we compute is such that:

$$\Pr_{C\sim 2^A}[\sum_{i\in C} \varphi_i(\hat{v}) \geq v(C) - \hat{e}] = \Pr_{C\sim 2^A}[\text{sign}_{w_{\hat{e}}}([1, \sum_{i\in C} \varphi_i(\hat{v}) - v(C)]) = 1] \geq 1 - \delta$$

$\square$

Lastly, we remark that for games whose characteristic functions are CGA models of order higher than 2, the Shapley is not the Nucleolus. An interesting extension of this work could be developing faster, sample-based methods for computing the Least Core with higher order CGA models.

| L2 regularization/$\hat{V}$ Rank | 1 | 2 | 5 | 10 | 20 | 35 |
|---|---|---|---|---|---|---|
| 0.001 | 0.256 | 0.092 | **0.066** | 0.067 | 0.068 | 0.069 |
| 0.01 | 0.261 | 0.104 | 0.091 | 0.090 | 0.093 | 0.090 |
| 0.1 | 0.679 | 0.679 | 0.652 | 0.646 | 0.669 | 0.664 |

Table 1: Results of hyper-parameter sweep for the second order CGA in the OpenAI particle world experiment. MSE is shown, lower is better.

| L2 regularization/$\hat{V}$ rank | 5 | 10 | 20 | 50 | 100 |
|---|---|---|---|---|---|
| 0.001 | 0.61 | 0.6214 | 0.6086 | 0.5971 | 0.5929 |
| 0.01 | 0.64 | 0.6414 | 0.6414 | **0.6429** | 0.6429 |
| 0.1 | 0.6214 | 0.63 | 0.62 | 0.6286 | 0.6242 |

Table 2: Results of hyper-parameter sweep for the second order CGA in the NBA data. Model accuracy is shown, higher is better.

## A.8 Experiments Hyper Parameter Search

In the low rank approximations of $\widehat{V}$ (as suggested by [28, 38]), we represented a team $C$ via a one-hot encoding $\boldsymbol{x_C}$ and fit a model of the form:

$$\hat{v}(C) = \boldsymbol{w}^T \boldsymbol{x_C} + \boldsymbol{x_C}^T \hat{V} \boldsymbol{x_C}$$

We tried parameterizing $\hat{V}$ via a low-rank matrix and swept weight decay ($l_2$ regularization) parameters on our validation set. Here we report the results of the full sweep for both of our experiments.

Table 1 shows the MSE (lower is better) of the performance prediction for various parameter values in the OpenAI particle world experiment. Table A.8 shows the accuracy of the model in predicting wins (higher is better) in the NBA experiment. In both cases we see that a relatively low rank model does very well at capturing structure in our environments. The main text analyzes the models resulting from these parameter choices.

# B  Additional Work in Cooperative Game Theory

In computational, cooperative game theory [8], there has been ample, albeit orthogonal prior work that studies CF representations and Shapley computation. We may classify them as follows:

- Fast methods to compute the SV for certain subclasses of games: [14] (for voting games). By contrast, our representation permits facile Shapley computation for all cooperative games.

- Sample complexity of approximating the SV: [32], [1] (only for simple games), [29] (only for supermodular games). By contrast, our bounds focus on the sample complexity of learning the CF function with the CGA representation, thus drawing upon PAC/PMAC techniques. None of the other works in this category need to nor use learning theoretic methods. They focus only on studying the concentration of estimated SV values via standard concentration inequalities.

- Representation designed to allow for easy computation of the SV: [33] (only for networks), [11]. By contrast, our CGA model has provable, learning theoretic properties (and additionally, practical success on real world data). The provable guarantee is crucial since we need to use the model to learn the unknown CF from data.