[Reviews · NeurIPS 2020]

Review 1

Summary and Contributions: The paper makes two interesting contributions to the cooperative game theory literature. First, the paper illustrates how to provide principled estimations of characteristic functions using parametric models while leveraging lossy abstractions. Second, the paper determines the relationship between error in the characteristic function error and error in the Shapley value. Experiments show how the proposed approaches can be used.

Strengths: Overall, I really liked this paper. To the best of my knowledge (and a bit to my surprise) this is the first work that uses lossy abstractions for characteristic functions, the relationship between the abstraction and Shapley value error was interesting and useful to characterize, the experiments were reasonable, and the paper was clearly written. The topic is a reasonable fit for the conference given its focus on learning and game-theoretic foundations. Furthermore, I think the results will have broader interest to the community as the Shapley value gains in interest.

Weaknesses: I thought there were some weaknesses in the description of some of the experimental results. In Figure 1, while I can see some structure on the diagonal, I would have appreciated clearer discussion as to what other structure I might be looking for. For the correlations (Fig 2) I would have appreciated a more detailed discussion of the data instead of just "eye-balling" the graphs. It might also have been interesting to add some more commentary to practitioners about choosing the order of the model - for example a commentary on classes of games and waht order k model they map well to.

Correctness: The claims appeared to be correct. The experiments seemed to be sound with respect to their design.

Clarity: The paper was well written. Some clarification is required with respect to Figure 1.

Relation to Prior Work: There has been work on approximating the Shapley value if the characteristic function is known (for example, for voting games -- Fatima et al (2008), A linear approximation method for the Shapley value, or Y. Bachrach, J. Rosenschein, (2008) Approximating power indices, and probably others). These are a little orthogonal to the work presented here in that they focusing on Shapley value approximations, but might be of interest to the authors. Owen (1972), Multilinear extensions of games

Reproducibility: Yes

Additional Feedback: I have read the authors' response. I continue to think this was an interesting paper.


Review 2

Summary and Contributions: This paper deals primarily with estimating characteristic functions from data using the notion called cooperative game abstractions (CGAs). This is an important problem to address as the specific form of characteristic function CF is often not known and also computing Shapley values is a combinatorial problem. The authors provided identification results, sample complexity bounds for CGAs, and also error bounds in estimating Shapley values using this framework.

Strengths: 1. This work is first to use abstraction for computational tractability in the space of cooperative games; 2. Shapley Value can be computed from a simple weighted sum of the CGA parameters; 3. Solid theoretical results to back the proposed framework.

Weaknesses: 1. There is no systematic way for the practitioners to understand which order CGAs are suitable to their application context 2. Proof sketches should have been given in the main stream paper (at least for one or two major results).

Correctness: The proposed claims as well as method seem to be correct.

Clarity: Yes, the paper is well written and easy to follow.

Relation to Prior Work: Yes, the authors did a great work in distinguishing their work from that of the literature.

Reproducibility: Yes

Additional Feedback: Below are a few more comments on this paper: (a) Can you provide a systematic method that makes the job of practitioners simple to understand the suitable order CGAs to use in their application context? 2. Please provide the outline of the proofs for a couple of key results in the main stream paper. Clearly, space is a problem. However, without the formal proofs in the main stream paper, even the paper is also not complete. 3. What are time and space complexities of the proposed method? 4. What happens if we increase the team size to a reasonable value such as 100 players? Does the proposed method scale to this setting in practice?


Review 3

Summary and Contributions: The paper proposes a method for approximating a characteristic function of a cooperative (coalitional game) and approximating the Shapley value. A characteristic function captures the values of every subset of agents can achieve in a (transferable utility) coalitional game, but in many cases it is not given directly. Similarly the Shapley value averages the marginal contribution over all agent permutations, and is computationally hard. The authors propose using a parametric model called CGA that estimates these from data, and allow for a linear time approximation of the Shapley value. The author give some complexity bounds and examine the method empirically on professional sport data.

Strengths: The work deals with the important and sometimes neglected issue of using learning techniques to solve coalitional (cooperative) games. The issue or representing and solving cooperative games has been given significant attention in the AI literature, as this is the key model of teamwork. However, these typically take the form of a restriction on the structure of synergies in the game, and this work uses learning techniques which is interesting.

Weaknesses: I believe the results proposed in this paper are related to existing work. The techniques used are close to existing methods - at the very least a detailed comparison is in order. The paper fails to acknowledge lots of literature on representing coalitional games in a restricted manner. In fact, many techniques have been proposed for concisely representing coalitional games, and approximately solving them. This issue is covered in depth in (e.g): Chalkiadakis, Georgios, Edith Elkind, and Michael Wooldridge. "Computational aspects of cooperative game theory." Synthesis Lectures on Artificial Intelligence and Machine Learning 5.6 (2011): 1-168. Michalak, Tomasz P., et al. "Efficient computation of the Shapley value for game-theoretic network centrality." Journal of Artificial Intelligence Research 46 (2013): 607-650. Conitzer, Vincent, and Tuomas Sandholm. "Computing Shapley values, manipulating value division schemes, and checking core membership in multi-issue domains." AAAI. Vol. 4. 2004. Conitzer, Vincent, and Tuomas Sandholm. "Complexity of constructing solutions in the core based on synergies among coalitions." Artificial Intelligence 170.6-7 (2006): 607-619. There are many good approximations for the Shapley value, including: Fatima, Shaheen S., Michael Wooldridge, and Nicholas R. Jennings. "A linear approximation method for the Shapley value." Artificial Intelligence 172.14 (2008): 1673-1699. The issue of proving a lower bound on the number of queries required to learn a characteristic function has been also investigate earlier (some of the techniques in these are similar to what is proposed in this paper): Maleki, Sasan, et al. "Bounding the estimation error of sampling-based Shapley value approximation." arXiv preprint arXiv:1306.4265 (2013). Bachrach, Yoram, et al. "Approximating power indices: theoretical and empirical analysis." Autonomous Agents and Multi-Agent Systems 20.2 (2010): 105-122. Liben-Nowell, David, et al. "Computing shapley value in supermodular coalitional games." International Computing and Combinatorics Conference. Springer, Berlin, Heidelberg, 2012. I think the topic is very interesting, but given the above work, I think the innovation is limited. Also, the empirical evaluation is carried on only one dataset, and a more thorugh coverage is warrented. Additionally, this looks to me like a game theory paper with some learning aspects, not a machine learning paper.

Correctness: Yes. However, you need to more clearly state the assumptions you make regarding the game, and where you use them.

Clarity: The paper is reasonably well written, but you need to better motivate the choice of data.

Relation to Prior Work: No, a much more detailed comparison is in order.

Reproducibility: Yes

Additional Feedback: This is a good starting point, but you need to better clarify how this work differs from exisiting work, and the advantages of these approaches. I think that there are many reasonable restrictions on the strcuture of the characteristic function that allow for better approximations of the Shapley value. Also, there are many other solutions - why do you focus on Shapley? Or do the methods generalize to other solutions as well? What do the machine learning community learn from this work? Does the ML community typically look at coalitional games? Maybe teams in multiagent reinforcement learning can be characterized like this? You have to make the impact more clear. Post feedback: Following the discussion, I think this work is interesting and with some novel elements. Again, there is already a lot of literature on this (more on the game theory side than the ML side) - I suggest giving this the additional though it warrants.


Review 4

Summary and Contributions: The paper gives a family of approximations to the characteristic function of a cooperative game and shows how the parameters of these approximations can be estimated and used to efficiently calculate an estimate of the Shapley value of the participating players. It gives necessary and sufficient conditions for identifying the correct model and relates errors in the estimates to errors in the resulting Shapley values. Finally, it demonstrates the usefulness of the model on a multi-agent RL domain and on real-world Basketball data.

Strengths: - Clear objective of the paper, addressing an interesting and important problem in multi-agent RL and team composition - Clean and sound modelling approach, with the appealing property that the approximate Shapley values can be calculated efficiently from the approximation's parameters - Good theoretical results about identification and errors - Great experiments showing how the model works in practice - both on multi-agent RL and sports data. - Work makes instruments of cooperative game theory more accessible to ML community

Weaknesses: - Generalisation to k-fold interactions seems not relevant in practice (according to their own experiments), but k=2 case was already known. - Possibly missing some related work in ranking/skill estimation/matchmaking, e.g., TrueSkill (which can estimate player skills from team outcomes, but does not use pairwise or higher order interactions.)

Correctness: Claims seem plausible, but I was not able to check the proofs.

Clarity: Very well written paper.

Relation to Prior Work: Related work sections seems to be missing some important references, including: Chalkiadakis, Georgios, Edith Elkind, and Michael Wooldridge. "Computational aspects of cooperative game theory." But also related work by Conitzer and Sandholm, by Bachrach et al, by Shoham et al. In light of this, the novelty of this work may be limited.

Reproducibility: Yes

Additional Feedback: Very nice paper, but needs some work on the analysis of the existing literature and context.

[Author Response · NeurIPS 2020]

We thank the reviewers for the detailed feedback! We will be sure to clarify the figures and the text as well as incorporate
proof sketches as suggested. In the interest of space, we group and address the main concerns below:

**Reviewer 1:**

*"[on] choosing the order of the model"* + **Reviewer 2's** *"[what is a] systematic way for the practitioners to understand*
*which order CGAs are suitable to their application context"* — The order $k$ of the CGA model is dependent on the
application at hand. It may be set to be the maximum $k$-way interaction that the practitioner expects to take place in
the team. Our model is especially useful in settings like the NBA, in which team sizes are small relative to the overall
number of players. This structural prior can be encoded in the order of the CGA. As an example, for the NBA, we
expect at most $5$-way interaction and so it is necessary to only consider compact, low-rank models.

**Reviewer 2 (please see the above response to Reviewer 1 also):**

*"time and space complexities"* — The time to compute the SV is the space complexity of the CGA model: the number of
parameters. The complexity of learning a CGA depends on the training method employed to learn $\hat{v}$ (e.g. we use SGD).

*"What happens if we increase the team size to a reasonable value such as 100 players? Does the proposed method scale "*
— We believe our method would scale well with the team size, and do even better if the data exhibits low-rank structure.
Note that we considered doing this larger-scale experiment, but were limited by the speed of current MARL methods.
Training $\binom{12}{3}$ teams already takes $4$ days, so training $\binom{N}{100}$ teams for $N$ much larger may well take on orders of months.

**Reviewer 3:**

*"[How are] results proposed in this paper are related to existing work"* + **Reviewer 4's** *"missing some important*
*references...[thus] the novelty of this work may be limited* — Broadly, there has been ample, but more distantly related
work that fall into the following three categories with which we characterize all the papers mentioned:

1) Fast methods to compute the SV (usually for certain subclasses of games): Fatima et al (only for voting games).

2) Sample complexity of approximating the SV: Maleki et al; Bachrach et al (only for simple games); Liben-Nowell et
al (only for supermodular games).

3) Representation designed to allow for easy computation of the SV: Michalak et al (only for networks); Conitzer et al
'04 (the '06 paper focuses on the core); Shoham et al (already cited).

To the best of our knowledge, and as also noted by Reviewer 1, all prior work in these three categories assume full
knowledge of the CF. By contrast, our central premise is – \*\*the CF is unknown and needs to be learnt from data\*\*. We
describe how our work differs from each of the three categories below and will be sure to append this to the paper:

1) our method holds for all cooperative games, to answer the question on *"assumptions you make regarding the game"*.

2) our bounds are on the sample complexity of learning the CF function with the CGA representation, thus drawing
upon PAC/PMAC techniques. In contrast, work in (2) focus on the concentration of estimated SV values via standard
concentration inequalities like Chebyshev/Hoeffding, and do not need to nor use learning theoretic methods. Thus, we
respectfully disagree with that *"the techniques in these are similar to what is proposed in this paper"*.

3) we propose a representation that not only allows for fast SV computation, but also has provably good learning
theoretic properties (crucial since we use it to learn the unknown CF from data). CGA is the first model to met not only
the first desideratum (like work in (3)), but the second as well. We believe we are the \*\*first to design a representation
with learning from samples in mind\*\*. Thus, we think there is ample and *"not limited"* novelty to our work.

*"[how are]... teams in multiagent reinforcement learning... characterized"* — We actually perform exactly this empirical
evaluation in a large-scale MARL setting, where we exhaustively evaluate all counterfactual teams to characterize what
a CF would look like. Please note also that we evaluate on two large scale settings and not one, as you stated.

**Reviewer 4 (please see the first response to Reviewer 3 also):**

*"Generalisation to k-fold interactions seems not relevant in practice (according to their own experiments), but k=2 case*
*was already known."* — To clarify, while there has been work that experiments with order $k = 2$ models, no prior work
has considered nor compared it with higher orders models. So it is not "already known" if second order models are
indeed the best or offer the best tradeoff. Our experiments are the first to do precisely this comparison. Also, we found
that the third order CGA doe perform better, but not to the extent that is worth the space and runtime tradeoff of $O(n^3)$
vs $O(n^2)$. And so, we think our comprehensive comparison and its conclusion is a novel contribution.

*"missing ... ranking/skill estimation/matchmaking [models like] e.g., TrueSkill "* — Note that all these models fall under
the umbrella of the CGA family; it is a complete representation (Fact 1). For instance, TrueSkill is a first order CGA,
which we include in our experiments, since the team performance is set to be the weighted sum of its players' skills.

[Meta-Review · NeurIPS 2020]

Three of four reviewers felt the paper should be accepted. The fourth reviewer (R3) has some reasonable points that the authors should to clarify their contribution, but I believe the review is also a little overly harsh (and the reviewer also acknowledged that). The author response appears to effectively address many of R3's concerns. I concur with accepting the paper, recommending that the authors thoroughly address the reviewers' comments in the final version.